# G-CSF drives autoinflammation in APLAID

Elisabeth Mulazzani[1,2], Klara Kong[1], Juan I. Aróstegui [3], Ashley P. Ng [4,5,6], Nishika Ranathunga[2,7], Waruni Abeysekera [2,7], Alexandra L. Garnham[2,7], Sze-Ling Ng[8], Paul J. Baker[1,2], Jacob T. Jackson [9], John D. Lich[8], Margaret L. Hibbs[10], Ian P. Wicks [1,2,11], Cynthia Louis [1] & Seth L. Masters [1,2] ✉

Missense mutations in *PLCG2* can cause autoinflammation with phospholipase C gamma 2-associated antibody deficiency and immune dysregulation (APLAID). Here, we generated a mouse model carrying an APLAID mutation (p.Ser707Tyr) and found that inflammatory infiltrates in the skin and lungs were only partially ameliorated by removing inflammasome function via the deletion of caspase-1. Also, deleting interleukin-6 or tumor necrosis factor did not fully prevent APLAID mutant mice from autoinflammation. Overall, these findings are in accordance with the poor response individuals with APLAID have to treatments that block interleukin-1, JAK1/2 or tumor necrosis factor. Cytokine analysis revealed increased granulocyte colony-stimulating factor (G-CSF) levels as the most distinct feature in mice and individuals with APLAID. Remarkably, treatment with a G-CSF antibody completely reversed established disease in APLAID mice. Furthermore, excessive myelopoiesis was normalized and lymphocyte numbers rebounded. APLAID mice were also fully rescued by bone marrow transplantation from healthy donors, associated with reduced G-CSF production, predominantly from non-hematopoietic cells. In summary, we identify APLAID as a G-CSF-driven autoinflammatory disease, for which targeted therapy is feasible.

APLAID is a rare, autoinflammatory syndrome characterized by both systemic inflammation and mild to severe immunodeficiency[1]. Clinical and laboratory features of APLAID include recurrent blistering skin lesions, pulmonary manifestations, joint pain, inflammatory eye and bowel disease, accompanied by a reduced number of class-switched memory B cells and recurrent bacterial infections[1]. Disease onset occurs typically during infancy or early childhood with varying degrees of severity. A total of ten clinical cases of APLAID have been reported (Extended Data Table 1)[1–6]. To date, there is no effective therapy for APLAID and treatment with tumor necrosis factor (TNF) inhibitors

and the interleukin (IL)-1 receptor antagonist anakinra only partially improved the disease course[5].

APLAID is a dominantly inherited disease caused by monoallelic missense mutations in the *PLCG2* gene[1]. Aside from APLAID, other inherited *PLCG2* mutations are identified in phospholipase C gamma 2 (PLCγ2)-associated antibody deficiency and immune dysregulation (PLAID)[7], while acquired mutations and variants have been reported in cancer[8] and neurodegenerative diseases[9].

PLCγ2 is highly conserved among species and characterized by a multidomain insert between the X Box and Y Box, which consists

[1]Inflammation Division, The Walter and Eliza Hall Institute of Medical Research, Parkville, Victoria, Australia. [2]Department of Medical Biology, University of Melbourne, Parkville, Victoria, Australia. [3]Department of Immunology, Hospital Clínic-IDIBAPS, Barcelona, Spain. [4]Blood Cells and Blood Cancer Division, The Walter and Eliza Hall Institute of Medical Research, Parkville, Victoria, Australia. [5]Clinical Haematology Department, Royal Melbourne Hospital, Melbourne, Victoria, Australia. [6]Peter MacCallum Cancer Centre, Parkville, Victoria, Australia. [7]Division of Bioinformatics, The Walter and Eliza Hall Institute of Medical Research, Parkville, Victoria, Australia. [8]Immunology Research Unit, GlaxoSmithKline, Collegeville, PA, USA. [9]Division of Immunology, The Walter and Eliza Hall Institute of Medical Research, Parkville, Victoria, Australia. [10]Department of Immunology and Pathology, Monash University, Clayton, Victoria, Australia. [11]Rheumatology Unit, Royal Melbourne Hospital, Parkville, Victoria, Australia. ✉e-mail: masters@wehi.edu.au

of a split PH domain, N-terminal SH2 (nSH2) domain, C-terminal SH (cSH2) domain and an SH3 domain. The two interaction surfaces (the split PH/catalytic domain and the cSH2/C2 domain) keep PLCγ2 in an autoinhibited form[10]. Among the reported APLAID cases, a total of six different mutations have been identified, almost all located within the regulatory region, resulting in failure of autoinhibition, constitutive phospholipase activity and an increased production of both intracellular inositol-1,4,5-trisphosphate (IP$_3$) and calcium[1].

PLCγ2 is triggered upon activation and phosphorylation of non-receptor tyrosine kinases (such as SYK) or Tec kinases (such as BTK)[11], resulting in phosphorylation of PLCγ2 at multiple sites[12]. In turn, PLCγ2 converts phospholipid phosphatidylinositol-4,5-bisphosphate (PIP$_2$) into the second messengers diacylglycerol (DAG) and IP$_3$[13], resulting in the release of endoplasmic reticulum-stored calcium. The role of calcium as a second messenger is well established ranging from stimulation of cell proliferation and cell growth to lymphocyte activation[14]. However, it is the hematopoietic cell type that determines the consequence of PLCγ2 activation. In the context of APLAID, impaired B cell differentiation and enhanced myelopoiesis are the key immunological features (Extended Data Table 1), which can be explained by the critical role of PLCγ2 in both cell types. While B cell receptor signaling requires the cSH2 and C2 domain interfaces of PLCγ2 to associate with BLNK and the B cell signalosome[15], thus affecting survival of mature B cells and antibody production, myeloid cells also depend on PLCγ2 for myeloid differentiation and hematopoietic development[16,17].

The potential mechanism by which autoinflammation is promoted in APLAID remains elusive. In vitro studies have implicated the NLRP3 inflammasome as patients' peripheral blood mononuclear cells secreted increased levels of IL-1β in response to lipopolysaccharide priming alone, and this effect was attenuated by using a PLC inhibitor, intracellular calcium blockers or an adenylate cyclase activator[18]. Others found that PLCγ2 variants activate the NLRP3 inflammasome through the noncanonical pathway that requires ATP or nigericin as stimuli after lipopolysaccharide priming[5]. However, patients do not have a robust and durable response to blocking the inflammasome cytokine IL-1β[5].

*N*-ethyl-*N*-nitrosourea (ENU) mutagenesis studies have created mice with spontaneous mutations in *Plcg2*. The initially described model was the *Ali5* mouse, which carried a heterozygous gain-of-function mutation in *Plcg2* at p.Asp993Gly[19], leading to severe spontaneous inflammation and autoimmunity. The second mouse model, *Ali14*, bore a heterozygous gain-of-function mutation at p.Tyr495Cys of the split PH domain of *Plcg2*, and exhibited spontaneous hind paw swelling/inflammation, hypergammaglobulinemia and infertility[20]. Although these models improved our mechanistic understanding, further in vivo studies were hampered because the phenotype did not transfer to the C57BL/6 background.

To dissect the pathogenicity of APLAID disease in an in vivo setting and to identify therapeutic regimes, we generated a de novo mouse model, which carries the exact human p.Ser707Tyr mutation as found in APLAID.

## Results

### Phenotype of mice encoding an APLAID mutation in *Plcγ2* (p.Ser707Tyr)

We generated C57BL/6 mice in which the APLAID mutation p.Ser707Tyr was introduced into the *Plcg2* locus via targeting methodology, after a Neomycin cassette flanked by Frt sites that prevents expression of the mutant allele until Flp was introduced into the system (Fig. 1a). Heterozygous *Plcg2* offspring were then bred with C57BL/6 Flp deleter mice, which allowed the excision of the Neomycin stop selection cassette to obtain *Plcg2*[S707Y/+] mice. Phenotypically, *Plcg2*[S707Y/+] mice display dry, flaky skin at birth. As disease progresses, skin inflammation develops on the paws, ears and tail (Fig. 1b). Post-weaning severity of skin lesions was assessed by a clinical score ranging from 0 (no clinical signs) to 5 (clinical endpoint; Fig. 1c and Methods). *Plcg2*[S707Y/+] pups exhibited lower body weight over time compared to littermate controls (Fig. 1d) and developed splenomegaly (Fig. 1e). Computed tomography (CT) scans revealed reduced bone mass compared to littermate controls (Extended Data Fig. 1a). Lifespan was dramatically shortened in *Plcg2*[S707Y/+] mice, which typically died within 6 weeks of age (Fig. 1f). Histologic examination of skin, lung and gut tissues revealed infiltrations of leukocytes (CD45+; Extended Data Fig. 1b). While T (CD3+Ki67) and B (B220+) cells remained unaltered (Extended Data Fig. 1b) in comparison to wild-type (WT) littermate controls (Extended Data Fig. 1c), neutrophils (MPO+) and macrophages (F4/80+) were found to be the dominant immune cell type (Fig. 1g) in the different analyzed organs of the APLAID disease model. Notably, neutrophils (MPO+) resided rather in the tissue than in the blood vessel endothelium (Extended Data Fig. 1d). Automated peripheral blood cell analysis (ADVIA) showed a slight increase of neutrophils and a mild decrease in lymphocytes during disease onset compared to controls (Fig. 1h). There was no difference in platelet and red blood cell counts between *Plcg2*[S707/+] mice and controls (Extended Data Fig. 1e–g). Fluorescence-activated cell sorting (FACS) analysis of the skin revealed an expansion of monocytes/macrophages and increased neutrophil counts in the lung during disease onset compared to *Plcg2*[+/+] mice, while neutrophil counts were decreased in the bone marrow (BM; Fig. 1i,j). At peak of disease (approximately 6 weeks of age), skin and lung of APLAID mice showed a notable neutrophil infiltration compared to *Plcg2*[+/+] mice, but no difference of neutrophil or myeloid cell counts in blood and BM was observed

**Fig. 1 | Phenotype of mice encoding an APLAID mutation in *Plcg2* (p.Ser707Tyr). a**, Scheme of Flp-excision at the *Plcg2* recombined locus. Breeding was established with C57BL/6 Flp deleter mice to excise the Neomycin selection cassette and to generate heterozygous mice carrying the Neo-excised point knock-in mutation p.Ser707Tyr. **b**, *Plcg2*[S707Y/+] mice displayed cutaneous lesions on paws, ears and tail. **c**, Severity of skin inflammation is reflected by an APLAID skin score of 2 on a 0–5 scale after weaning up to 24 d of age (*Plcg2*[+/+], *n* = 19; *Plcg2*[S707Y/+], *n* = 5; at 2–3 weeks of age). **d**, A growth curve exhibits stunted weight gain of *Plcg2*[S707Y/+] mice (*Plcg2*[+/+], *n* = 14; *Plcg2*[S707Y/+], *n* = 9; at 2–4 weeks of age). **e**, *Plcg2*[S707Y/+] mice (*n* = 5) showed splenomegaly compared to their *Plcg2*[+/+] littermate controls (*n* = 7) at 6 weeks of age. **f**, Kaplan–Meier analysis demonstrated survival rates of 6 weeks in *Plcg2*[S707Y/+] mice after weaning (*Plcg2*[+/+], *n* = 5; *Plcg2*[S707Y/+], *n* = 6; at 4–6 weeks of age). **g**, At 6 weeks of age, histopathological examination demonstrated myeloid immune cell infiltration in paws, ears, tails, lung, gut and spleen. One representative immunohistochemistry (IHC) section (MPO+ and F4/80+) from three independent experiments is shown. **h**, During disease onset, data obtained from ADVIA blood cell analyzer revealed an increase of neutrophils and a mild decrease in lymphocytes (*Plcg2*[+/+], *n* = 7; *Plcg2*[S707Y/+], *n* = 4; at 2 weeks of age). **i,j**, FACS analysis showed an increase of myeloid cell counts in the skin and lung at disease onset, while neutrophil numbers were reduced in the BM (*n* = 3 mice per genotype; at 2 weeks of age). **k**, During disease, peak neutrophil counts normalized in the blood as assessed by ADVIA analyzer (*Plcg2*[+/+], *n* = 5; *Plcg2*[S707Y/+], *n* = 6; at 6 weeks of age). **l,m**, FACS analysis of the skin and lung demonstrated increased numbers of myeloid cell numbers as APLAID progressed, while myeloid numbers in the BM remained unaltered (*n* = 3 mice per genotype; at 6 weeks of age). **n,o**, FACS analysis of lymphoid cells displayed normal T cells and reduced B cells (*n* = 3 mice per genotype; at 5 weeks of age). **p**, Quantification of immunoglobulin subtypes measured by ELISA revealed increased IgG2a levels of *Plcg2*[S707Y/+] mice (*n* = 3; at 6 weeks of age) in comparison to *Plcg2*[+/+] littermate controls (*n* = 8; at 6 weeks of age). Error bars represent the mean ± s.e.m. Statistical significance for skin score was determined by a two-way analysis of variance (ANOVA) with Bonferroni post-test correction. Statistical significance for the survival curve was determined by a Mantel–Cox test. Spleen weights, longitudinal weight data and cell numbers between two groups were determined by a two-sided unpaired Student's *t*-test.

(Fig. 1k–m). We identified comparable T cell numbers in thymus and spleen and slightly reduced B cell counts in the BM and spleens of *Plcg2*^S707Y/+ mice (Fig. 1m–o). Unlike participants, APLAID mice showed unaltered or reduced (IgG2a) IgG levels in plasma (Fig. 1p).

Collectively, these data indicate that the clinical phenotype of *Plcg2*^S707Y/+ mice recapitulates much of the human disease APLAID, which is characterized by neutrophilic skin and lung inflammation together with mild immunodeficiency. However, in contrast to

individuals with APLAID, *Plcg2*^S707/+ mice lack inflammatory eye diseases and a pronounced immunoglobulin reduction in the plasma.

## Minimal rescue of APLAID mice after deletion of IL-6, caspase-1 or TNF

Based on previous in vitro results, we examined whether autoinflammation in APLAID is driven by IL-6, inflammasome or TNF. Surprisingly, *Plcg2*^S707Y/+IL-6^−/−, *Plcg2*^S707Y/+caspase-1^−/− and *Plcg2*^S707Y/+TNF^−/− mice

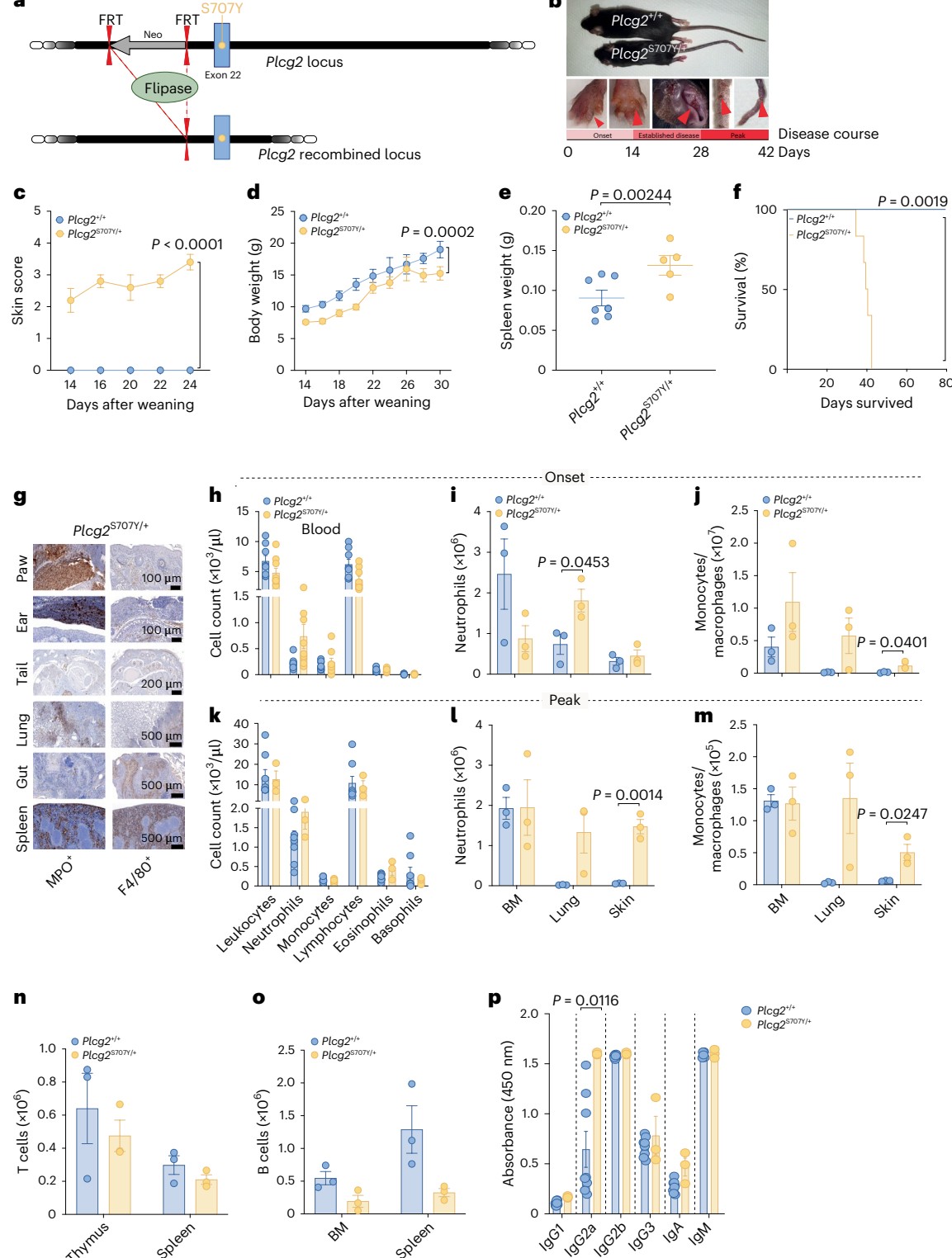

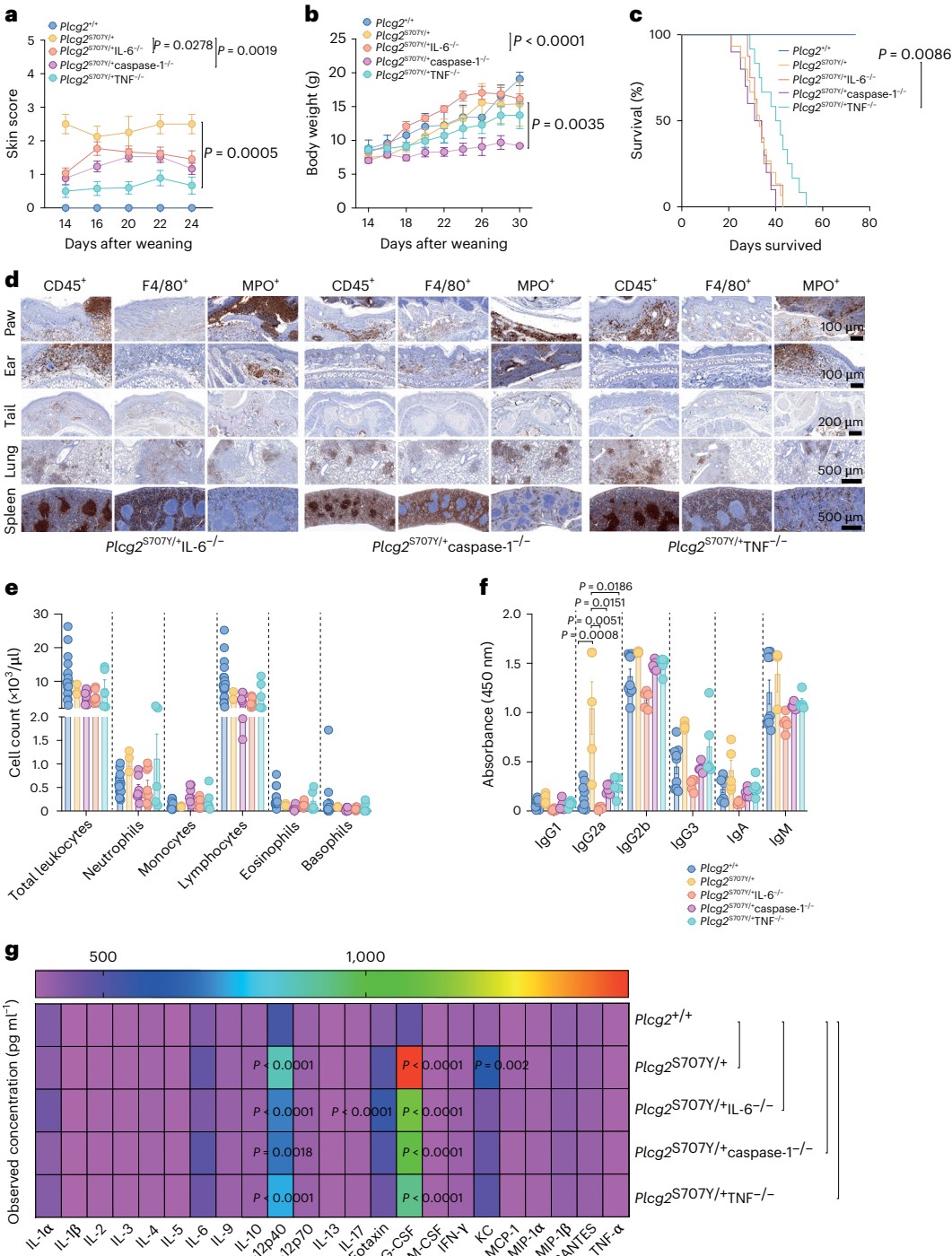

**Fig. 2 | Minimal rescue of APLAID due to deletion of IL-6, caspase-1 or TNF.**
**a**, A clinical APLAID skin score demonstrates persistent but attenuated skin inflammation of different *Plcg2*^S707Y/+ knockout mice after weaning (*Plcg2*^+/+, n = 19; *Plcg2*^S707Y/+, n = 4; *Plcg2*^S707Y/+IL-6^−/−, n = 26; *Plcg2*^S707Y/+caspase-1^−/−, n = 17; *Plcg2*^S707Y/+TNF^−/−, n = 20; at 2–3 weeks of age). **b**, A growth curve exhibiting stunted weight gain during weekly assessments among *Plcg2*^S707Y/+ knockouts (*Plcg2*^+/+, n = 4; *Plcg2*^S707Y/+, n = 5; *Plcg2*^S707Y/+IL-6^−/−, n = 5; *Plcg2*^S707Y/+caspase-1^−/−, n = 3; *Plcg2*^S707Y/+TNF^−/−, n = 4; at 2–4 weeks of age). **c**, Kaplan–Meier survival curve showing *Plcg2*^S707Y/+ knockout mice with an impaired survival rate in comparison to *Plcg2*^+/+ littermate controls after weaning (*Plcg2*^+/+, n = 10; *Plcg2*^S707Y/+, n = 15; *Plcg2*^S707Y/+IL-6^−/−, n = 8; *Plcg2*^S707Y/+caspase-1^−/−, n = 10; *Plcg2*^S707Y/+TNF^−/−, n = 12; at 3–7 weeks of age). **d**, Immune cell infiltrates were present in paws, ears, tails, lung, gut and spleen across all different mouse strains. One representative IHC section (from three independent experiments per genotype; at 4 weeks of age) of CD45^+, MPO^+ and F4/80^+ stains is shown. **e**, ADVIA analyzer data of the blood show variable neutrophil counts and a mild decrease of lymphocytes (*Plcg2*^+/+,

n = 15; *Plcg2*^S707Y/+, n = 3; *Plcg2*^S707Y/+IL-6^−/−, n = 8; *Plcg2*^S707Y/+caspase-1^−/−, n = 7; *Plcg2*^S707Y/+TNF^−/−, n = 5; at 4–5 weeks of age). **f**, Quantification of immunoglobulin subtypes measured by ELISA reveals abnormal levels similar to *Plcg2*^S707Y/+ mice (*Plcg2*^+/+, n = 9; *Plcg2*^S707Y/+, n = 5; *Plcg2*^S707Y/+IL-6^−/−, n = 5; *Plcg2*^S707Y/+caspase-1^−/−, n = 5; *Plcg2*^S707Y/+TNF^−/−, n = 5; at 4 weeks of age). **g**, Cytokine assessment by a multiplex assay in the plasma of *Plcg2* mice and different knockout mice demonstrates a large increase in G-CSF followed by IL-12p40 and eotaxin (n = 11 mice per genotype; at 4–6 weeks of age). Heat map colors represent mean cytokine values. Error bars represent the mean ± s.e.m. Statistical significance for skin score was determined by a two-way ANOVA for repeated measurements with Bonferroni post-test correction. Statistical significance for longitudinal weight data between two groups was determined by a two-sided paired *t*-test. The survival curves were analyzed by a Mantel–Cox test. Cytokine levels were established by an ordinary two-way ANOVA, and Bonferroni post-test correction was used to adjust for multiple testing.

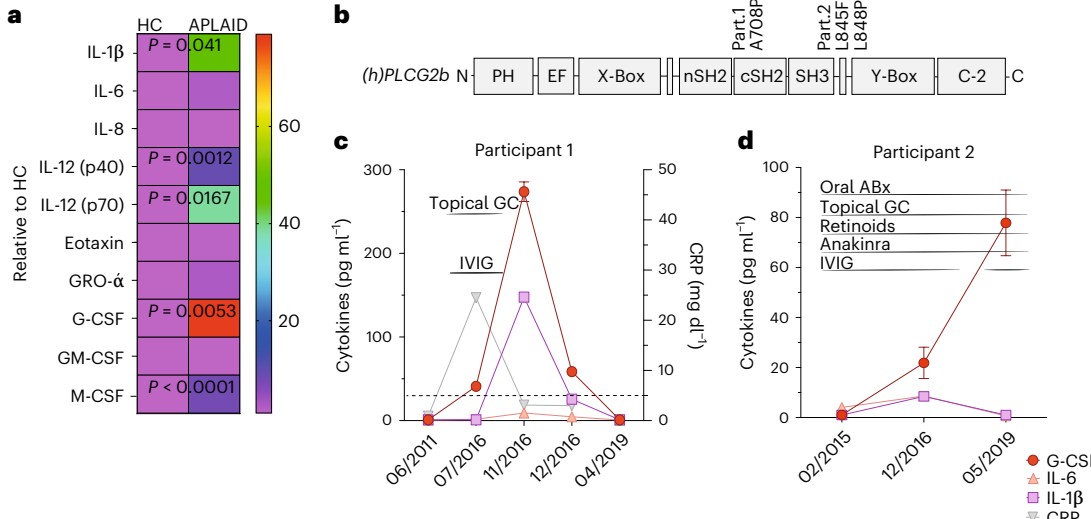

**Fig. 3 | High serum levels of G-CSF in participants with APLAID. a**, Cytokine assessment by a multiplex assay in sera of two APLAID participants showed a relative elevation in G-CSF followed by increased levels of IL-1ß, IL-12 and M-CSF compared to ten sex-matched and age-matched healthy controls (HCs). Color mapping of cytokine heat map represents mean values relative to control in pg ml⁻¹. **b**, Protein schematic and participant mutations in *PLCG2*. **c**,**d**, Longitudinal cytokine and CRP pattern of two APLAID participants (*n* = 2; Pat. 1 p.Ala708Pro/ male/13 years old; Pat. 2 p.Leu845Phe/848Pro/female/21 years old) under various treatment regimens revealed persistent elevation of G-CSF in participants' sera. Dashed line represents the cutoff for abnormal CRP values. Error bars represent the mean ± s.e.m. Cytokine levels between two groups were determined by an unpaired two-sided Student's *t*-test. ABx, antibiotics; CRP, C-reactive protein; GC, glucocorticoids; IVIG, intravenous immunoglobulins.

showed persistent skin inflammation as determined by the severity of the skin score (Fig. 2a). $Plcg2^{S707Y/+}IL-6^{-/-}$ mice had an increased weight gain compared to $Plcg2^{S707Y/+}$ mice (Fig. 2b). Post-weaning $Plcg2^{S707Y/+}caspase-1^{-/-}$ appeared runted and displayed a stunted weight gain compared to $Plcg2^{S707Y/+}$ mice (Fig. 2b). $Plcg2^{S707Y/+}TNF^{-/-}$ exhibited no difference with regards to body weight compared to $Plcg2^{S707Y/+}$ mice (Fig. 2b). Although $Plcg2^{S707Y/+}TNF^{-/-}$ mutants outlived $Plcg2^{S707Y/+}$ mice by about 12 d (Fig. 2c), $Plcg2^{S707Y/+}IL-6^{-/-}$, $Plcg2^{S707Y/+}TNF^{-/-}$ and $Plcg2^{S707Y/+}caspase-1^{-/-}$ mice all had decreased life expectancy compared to healthy littermate controls (Fig. 2c). Histologic examination revealed infiltration of leukocytes (CD45⁺), neutrophils (MPO⁺) and macrophages (F4/80⁺) into skin and lung (Fig. 2d), but not T and B cells (B220⁺; Extended Data Fig. 2a). ADVIA analysis showed a persistent neutrophilia in $Plcg2^{S707Y/+}TNF^{-/-}$ mice and a mild decrease in lymphocytes among all mutants and intercrossed mice (Fig. 2e). No major differences in platelets or red blood cell numbers were observed (Extended Data Fig. 2b–d). Abnormal IgG2a subtype levels were present as determined by ELISA (Fig. 2f). All clinical features of different knockout strains are summarized in Extended Data Table 2. Interestingly, a multiplex assay in the plasma of $Plcg2$ mice and $Plcg2$ mutant/ cytokine knockout mice revealed a marked increase in G-CSF levels and, to a lesser extent, the cytokines IL-12p40 and eotaxin (Fig. 2g). Other hematopoietic growth factors like macrophage colony-stimulating factor (M-CSF; Extended Data Fig. 2e) were detected neither in the supernatant of differentiated bone marrow-derived macrophages (BMDMs) from $Plcg2^{S707Y/+}$ mice nor in the plasma of $Plcg2^{S707Y/+}$ mice. Multiplex cytokine assessment of skin and lung lysates also revealed increased G-CSF and MIP-1α/MIP-1β in $Plcg2^{S707Y/+}$ mice, but not in $Plcg2^{S707Y/+}IL-6^{-/-}$, $Plcg2^{S707Y/+}TNF^{-/-}$ and $Plcg2^{S707Y/+}caspase-1^{-/-}$ mice (Extended Data Fig. 2f,g). In addition, we performed bulk RNA sequencing (RNA-seq) of neutrophils from $Plcg2^{S707Y/+}$ mice, which revealed a gene expression signature that closely resembles that of cells treated with G-CSF (Extended Data Fig. 2h)[21]. Taken together, autoinflammation in APLAID was not solely dependent on IL-6, inflammasome or TNF; however, G-CSF production was clearly dysregulated.

## G-CSF serum levels are high in individuals with APLAID

We examined if the elevated G-CSF levels in $Plcg2^{S707Y/+}$ mice are also seen in individuals with APLAID and found increased G-CSF, IL-1β and IL-12p70 in the sera of two treated individuals with APLAID (Fig. 3a)[5]. However, high G-CSF levels were not specific for APLAID and also present in individuals with other monogenic autoinflammatory disease (Extended Data Fig. 3). Participant 1 carries the heterozygous *PLCG2* gene mutation p.Ala708Pro (Fig. 3b) and presented with cutaneous lesions at disease onset and mild hypogammaglobulinemia followed by recurrent infections at later stages. Cutaneous flares were well controlled with topical steroids, and mild immunodeficiency was treated with intravenous immunoglobulins. Increased G-CSF levels correlated with increased C-reactive protein (CRP) and disease activity (Fig. 3c). Participant 2 has a compound heterozygous p.Leu845_Leu848 deletion in *PLCG2* (Fig. 3b) and clinically suffered from skin rash and recurrent lung bacterial infections leading to bronchiectasis and acute episodes of hemoptysis requiring partial pneumonectomy. Symptoms were inadequately controlled with anakinra, and there was persistent elevation of serum G-CSF (Fig. 3d). B cell immunodeficiency developed in early infancy and did not improve despite the continuous application of intravenous immunoglobulins. These data suggest that although the elevated G-CSF found in participants with APLAID fluctuates over time, it is not simply downstream of IL-1β activation, as observed in other conditions[22].

## Inhibition of G-CSF reverses established disease in *Plcg2*<sup>S707Y/+</sup> mice

The results from $Plcg2^{S707Y/+}$ mice and APLAID participants prompted us to assess the in vivo therapeutic potential of anti-G-CSF treatment. Monoclonal G-CSF antibody (50 µg) was injected intraperitoneally (i.p.) three times per week (Extended Data Fig. 4a) starting 14 d after birth, at which point mice had marked skin inflammation. A matched group of mutant mice received an appropriate IgG1 isotype control i.p. Remarkably, neutralization of G-CSF completely reversed established autoinflammatory disease, reflected by an APLAID skin score of 0 after 4 weeks of treatment (Fig. 4a). Moreover, $Plcg2^{S707Y/+}$ mice

that received anti-G-CSF injections displayed steady weight gain over time (Fig. 4b) and splenomegaly resolved (Fig. 4c). The lifespan of *Plcg2*^S707Y/+ mice following G-CSF antagonism was also prolonged (Fig. 4d). ADVIA analysis revealed reduced neutrophil and increased lymphocyte counts in *Plcg2*^S707Y/+ mice following anti-G-CSF injections (Fig. 4e). Importantly, neutrophil numbers of *Plcg2*^S707Y/+ mice did not drop below neutrophil counts of *Plcg2*^+/+ controls. Lymphocyte numbers also recovered in BM, but not in the spleen (Extended Data Fig. 4b), and immunoglobulin subtypes (Ig2a and IgM) decreased in *Plcg2*^S707Y/+ mice following anti-G-CSF treatment compared to isotype control (Fig. 4f). Thrombopoiesis and erythropoiesis remained largely unaltered in anti-G-CSF-treated *Plcg2*^S707Y/+ mice (Extended Data Fig. 4c–e) with the exception of the proportion of reticulocytes. FACS analysis of anti-G-CSF-treated *Plcg2*^S707Y/+ mice showed decreased total leukocytes and myeloid cells, particularly of neutrophils (CD45^+Ly6G^+CD11b^+) across all organs including the spleen, skin and lung (Fig. 4g–i). Histologic examination of isotype control-treated mice revealed persistence of immune cell infiltrates (H&E, MPO^+ and F4/80^+) in skin, lung and gut (Fig. 4j) compared to anti-G-CSF-treated mice (Fig. 4k). Thus, anti-G-CSF treatment of *Plcg2*^S707Y/+ mice rescued the clinical and laboratory features of established APLAID, improving weight gain, spleen size, lifespan and B cell abnormalities.

### Anti-G-CSF treatment supresses excessive myelopoiesis in *Plcg2*^S707Y/+ mice

Given the high levels of G-CSF together with myeloid cell accumulation in the BM, spleen, skin and lung, we next quantified myeloid progenitors and cell proliferation rates in *Plcg2*^S707Y/+ mice by a BrdU uptake assay. *Plcg2*^S707Y/+ mice had a marked increase in the proportion and numbers of neutrophils, monocytes and their progenitors as well as their proliferation rate in the BM (Fig. 5a–c). Notably, the enlarged spleens of *Plcg2*^S707Y/+ mice also harbored expanded numbers of neutrophils, monocytes and their progenitors and higher frequency of proliferating BrdU^+ cells (Fig. 5d–g), indicative of enhanced extramedullary myelopoiesis. In contrast, neutrophils, monocytes and macrophages in skin, lung and blood were all BrdU^− (Extended Data Fig. 5a–f), consistent with these cells being terminally differentiated in peripheral organs. Representative FACS plots showing the gating strategy were used to identify BrdU^+ neutrophils and monocytes in BM of a *Plcg2*^S707Y/+ mouse (Extended Data Fig. 5g).

A colony assay revealed that BM cells and splenocytes from APLAID mutant mice generated increased colonies particularly of the myeloid lineage, when cultured with cytokines such as G-CSF in vitro (Extended Data Fig. 6). In sum, excessive neutrophilia in *Plcg2*^S707Y/+ mice was driven by both BM and splenic extramedullary myelopoiesis.

To evaluate the importance of G-CSF in driving the enhanced myelopoiesis of *Plcg2*^S707Y/+ mice, we next investigated cell proliferation rates under anti-G-CSF therapy. Anti-G-CSF treatment reduced myeloid counts and decreased proliferation (BrdU^+ cells) in the BM (Fig. 5h–k)

and the spleen (Fig. 5l–o) relative to isotype control-treated APLAID mutant mice and to comparable levels observed in *Plcg2*^+/+ mice. These were consistent with the reduced disease activity in anti-G-CSF-treated *Plcg2*^S707Y/+ mice (Fig. 4). Representative flow cytometry plots showing cell numbers and individual frequencies of BrdU^+ cells from mice receiving anti-G-CSF antibody treatment or isotype control are shown (Fig. 5p). Taken together, these preclinical data indicate that antagonizing G-CSF prevents autoinflammation in APLAID mutant mice by suppressing medullary and extramedullary myelopoiesis.

### Bone marrow transplantation normalizes G-CSF and rescues APLAID phenotype

As indicated by quantitative PCR (qPCR; Extended Data Fig. 6a) and RNA-seq data, elevated G-CSF transcription was observed from both immune (myeloid) and non-immune (fibroblasts) cells from APLAID mice. To further address this, we attempted to rescue the *Plcg2*^S707Y/+ mediated autoinflammatory disease by transplanting lethally irradiated *Plcg2*^S707Y/+ mice with PLCG2^+/+ BM (Fig. 6a). Whole BM transplantation of WT cells diminished macroscopic skin inflammation in *Plcg2*^S707Y/+ mice (Fig. 6b), which was further underscored by a steady weight gain (Fig. 6c), normalized spleen size (Fig. 6d) and prolonged survival rates (Fig. 6e). ADVIA analysis revealed reduced neutrophil/monocyte counts and increased lymphocyte counts in *Plcg2*^S707Y/+ mice receiving WT BM cells (Fig. 6f). FACS analysis further confirmed decreased myeloid cell counts and no accumulation of neutrophils and monocyte/macrophages in the skin (Fig. 6g–i). BM chimeras exhibited normal amounts of G-CSF in plasma (Fig. 6j). Tissue pathology in *Plcg2*^S707Y/+ mice was prevented in BM chimeras with only a few immune cell infiltrates in skin and lung (Fig. 6k).

Importantly, irradiation alone did not improve signs of skin autoinflammation in *Plcg2*^S707Y/+ mice receiving BM from *Plcg2*^S707Y/+ mice (Extended Data Fig. 7b,c). Irradiated *Plcg2*^S707Y/+ mice still displayed weight loss (Extended Data Fig. 7d), splenomegaly (Extended Data Fig. 7e), reduced survival (Extended Data Fig. 7f) and neutrophilia (Extended Data Fig. 7g), while thrombopoiesis and erythropoiesis remained unaltered as determined by ADVIA analysis (Extended Data Fig. 7h–j). Donor WT CD45.1^+ cells fully reconstituted immune cells in the blood and spleen at 7 weeks after transplantation (Extended Data Fig. 7k) in *Plcg2*^S707Y/+ mice. FACS analysis of monocytes/macrophages confirmed a high degree of reconstitution with healthy WT donor cells in the lung, blood, spleen and BM, while there were less than 50% of skin macrophages (that is, Langerhans cells) in donor-derived WT CD45.1^+ cells (Extended Data Fig. 7k). This is consistent with the embryonic origin of Langerhans cells and their high resistance to irradiation[23].

These BM chimera experiments conclude that the host can still remain healthy with a *PLCG2* mutation in all non-hematopoietic cells. However, even though the *PLCG2* mutation is required in a hematopoietic cell, this does not establish if G-CSF is intrinsically coming from the same cell type.

---

**Fig. 4 | Blocking G-CSF reverses established disease in *Plcg2*^S707Y/+ mice.**
**a**, Post-weaning in vivo blockage of G-CSF completely reversed established autoinflammation in *Plcg2*^S707Y/+ mice, reflected by an APLAID skin score of 0 on a 0–5 scale after 4 weeks of treatment compared to *Plcg2*^S707Y/+ mice injected with an appropriate IgG1 isotype control. **b**, Growth curve exhibiting a steady weight gain of *Plcg2*^S707Y/+ mice following G-CSF blockage during weekly assessments. **c**, *Plcg2*^S707Y/+ mice treated with anti-G-CSF lacked an enlargement of the spleen compared to *Plcg2*^S707Y/+ mice receiving the isotype control. **d**, Kaplan–Meier analysis demonstrated exceeding survival rates of *Plcg2*^S707Y/+ mice following G-CSF antagonism in comparison with *Plcg2*^S707Y/+ mice treated with an IgG1 isotype control. **e**, ADVIA analyzer data of the blood revealed reduced neutrophil and increased lymphocyte counts in *Plcg2*^S707Y/+ mice after G-CSF neutralization. **f**, Quantification of immunoglobulin subtypes measured by ELISA revealed decreased IgG2a and IgM levels (n = 3 mice per genotype) in *Plcg2*^S707Y/+ mice

following anti-G-CSF treatment or isotype control. **g–i**, FACS analysis exhibited decreased numbers of total leukocytes and myeloid cells, particularly of neutrophils (Ly6G^+CD11b^+), across all organs including BM, spleen, skin and lung. **j,k**, Immune cell infiltrates were sparse in paws, ears, tails, lung, gut and spleen in anti-G-CSF-treated mice compared to isotype control. Representative IHC sections of HE, MPO^+ and F4/80^+ are shown. For all experiments, three mice per treatment group were used. All mice were 2 weeks old at the start of the experiment. Error bars represent the mean ± s.e.m. Statistical significance for skin score was determined by a two-way ANOVA. Statistical significance for longitudinal weight data was determined by a paired two-sided *t*-test. Spleen weights between two groups were determined by an unpaired two-sided Student's *t*-test. Statistical significance for the survival curve was determined by a Mantel–Cox test. Cell numbers between two groups were determined by an unpaired two-sided Student's *t*-test. mAB, monoclonal antibody.

## Delayed onset of APLAID phenotype in G-CSF-deficient mice

Next, we transferred $Plcg2^{S707Y/+}$ BM cells into either irradiated WT mice or G-CSF$^{-/-}$ mice (Fig. 7a) to assess if G-CSF originates from hematopoietic or non-hematopoietic cells in our APLAID disease model. As expected, transfer of BM cells from $Plcg2^{S707Y/+}$ mice resulted in inflammation as indicated by pronounced weight loss (Fig. 7b,c), reduced

survival (Fig. 7d) and elevated G-CSF (Fig. 7e) levels. In contrast, G-CSF knockouts showed a delayed onset of the APLAID phenotype (Fig. 7b–d). In accordance with the onset of clinical signs and body weight loss at day 70 following irradiation and reconstitution, increasing levels of G-CSF were observed in G-CSF$^{-/-}$ mice receiving $Plcg2^{S707Y/+}$ BM (Fig. 7e). Also, elevated IL-12 (p40) and eotaxin levels were observed

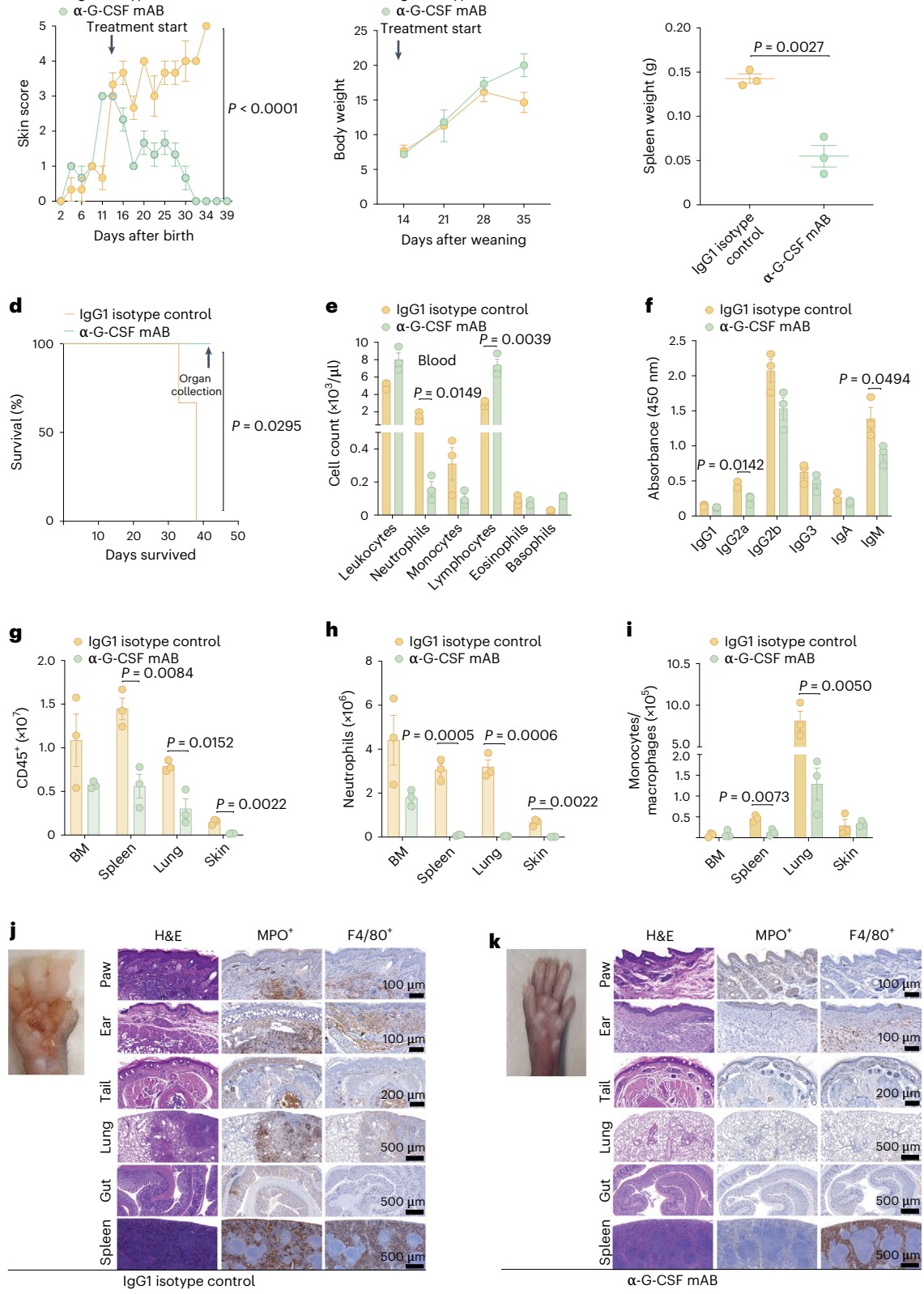

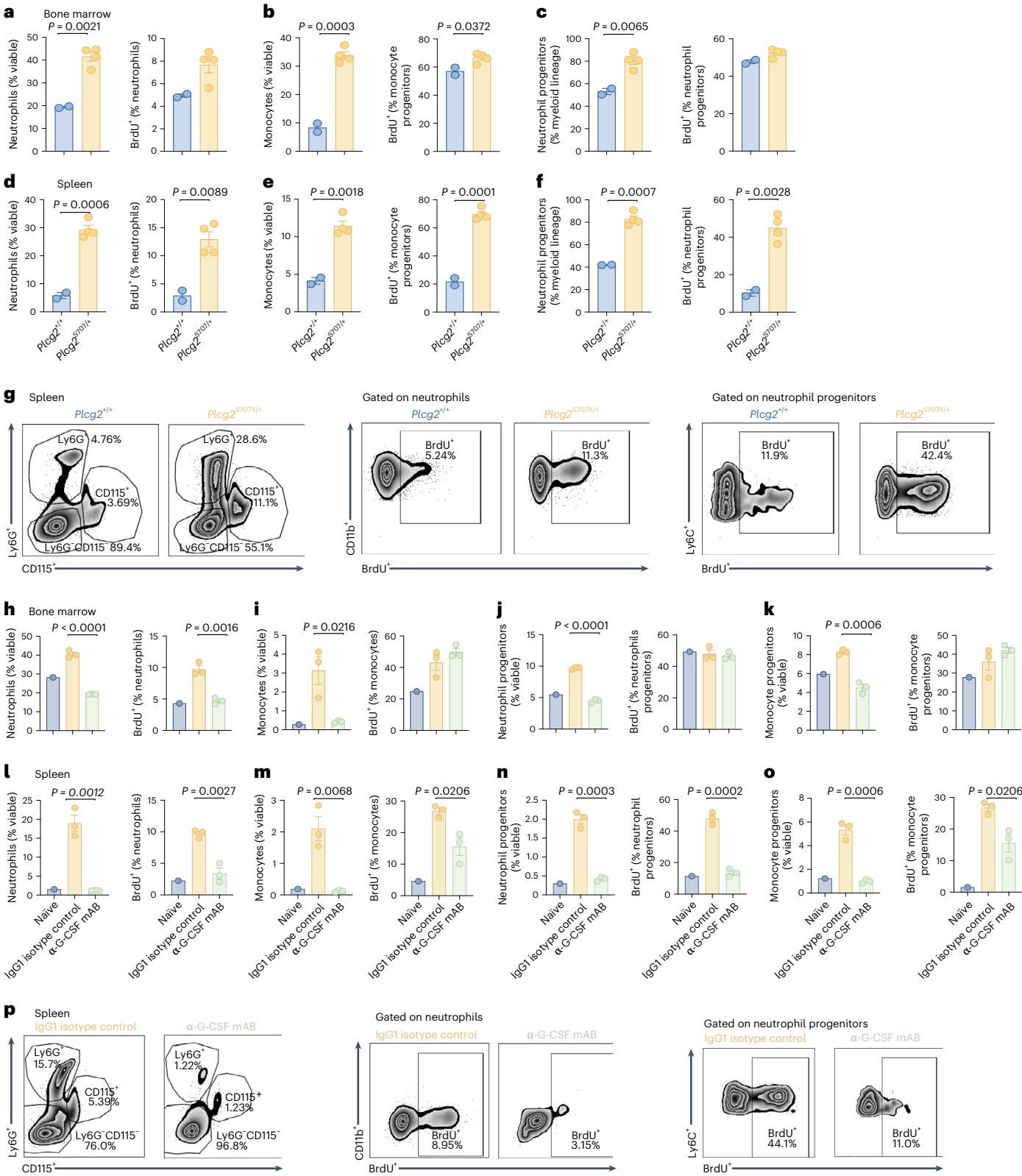

**Fig. 5 | Anti-G-CSF treatment suppresses enhanced myelopoiesis in APLAID mice. a–f**, *Plcg2*^S707Y/+ mice (*n* = 4 mice per group; yellow bar) had a marked increase in the proportion and numbers of neutrophils, monocytes and their progenitors as well as their proliferation rate in the BM and spleen compared to controls (*n* = 2 mice per group; blue bar). **g**, Representative FACS plots showing cell numbers and individual frequencies of BrdU⁺ cells. **h–o**, Anti-G-CSF treatment (*n* = 3 mice per group; green bar) reduced myeloid counts and

decreased proliferation (BrdU⁺ cells) in the BM and spleen relative to isotype control-treated APLAID mutant mice (*n* = 3 mice per group; yellow bar), and to a comparable level as that seen in *Plcg2*^+/+ mice (*n* = 1 mouse, blue bar). **p**, Representative FACS plots showing individual cell numbers and frequencies of BrdU⁺ cells under anti-G-CSF treatment or isotype control. Error bars represent the mean ± s.e.m. All mice were 4–5 weeks of age. Cell numbers between two groups were determined by an unpaired two-sided Student's *t*-test.

while animals are still asymptomatic (Extended Data Fig. 8a). Of note, the BM chimera with the highest G-CSF level was the one that had the most severe weight loss.

ADVIA analysis at day 18 showed APLAID-specific blood count abnormalities (increased neutrophils/monocytes and decreased numbers of lymphocytes; Fig. 7f) in irradiated and reconstituted WT mice, while thrombopoiesis and erythropoiesis (Fig. 7g,h) remained unaltered except for increased reticulocyte counts (Fig. 7i). Taken together, these findings suggest that G-CSF is predominantly derived from radio-resistant cells in our model, but it can eventually be made by hematopoietic cells too. Importantly, our data argue in favor of allogeneic BM transplantation or G-CSF neutralization as viable treatment options for individuals with APLAID.

## Discussion

Using genetic and pharmacological approaches, we identify G-CSF as the key driver of autoinflammation in APLAID caused by missense mutations in *PLCG2*. Allogeneic BM transplantation may be beneficial in individuals with APLAID as indicated by our preclinical dataset. Normalized G-CSF levels and reversal of tissue pathology in BM chimeras further underscore the pivotal role of this cytokine in perpetuating autoinflammation in APLAID. Moreover, the observation that *Plcg2*[S707Y/+] mice with established disease can be completely rescued by anti-G-CSF treatment, without inducing neutropenia, may offer a more appealing therapeutic strategy for individuals with APLAID in the future.

In the beginning, we undertook a genetic approach to define the molecular mechanisms and signaling pathways in APLAID. Although IL-6, inflammasome and TNF signaling have been implicated by previous in vitro studies[18], our work demonstrates that disease in *Plcg2*[S707Y/+] mice is largely independent of these factors. In contrast, we identify a specific role for enhanced G-CSF signaling in causing autoinflammatory disease in *Plcg2*[S707Y/+] mice. Although G-CSF levels are profoundly elevated in APLAID mice, this increase was not specific to APLAID participants compared to other monogenic autoinflammatory diseases. However, G-CSF levels in individuals with APLAID remained elevated despite the administration of biologics targeting IL-1, for example, arguing that G-CSF is not a nonspecific feature of inflammation in APLAID. Furthermore, our results explain the inadequate treatment responses to IL-1 or TNF blockade in individuals with APLAID.

Increased G-CSF levels across various autoinflammatory diseases levels may be the result of a shared organ pathology characterized by predominantly neutrophilic infiltrates. However, it is unknown if anti-G-CSF therapy might also be beneficial in hyper-IgD with periodic fever syndrome, deficiency of adenosine deaminase-2 or other autoinflammatory conditions. Although neutralization of G-CSF or its receptor has shown to be a promising therapeutic target in inflammatory joint diseases such as rheumatoid arthritis[24,25] and other autoimmune

disorders[26,27], we are unaware of preclinical studies or patient reports in autoinflammatory diseases. Our preclinical observations alongside G-CSF elevation in other autoinflammatory diseases implicate that targeting G-CSF can be considered in a variety of neutrophil-associated, monogenetically inherited human inflammatory diseases and could determine whether G-CSF is the primary driver in these autoinflammatory conditions or merely a disease modifier in a complex inflammatory cascade.

Importantly, defects in lymphopoiesis were restored in *Plcg2*[S707Y/+] mice receiving anti-G-CSF treatment. A possible explanation is that elevated G-CSF can suppress lymphopoiesis by targeting stromal cells that contribute to lymphoid niches in the BM[28]. This is a relevant consideration in determining the benefit of anti-G-CSF treatment for individuals with APLAID, where immunosuppression of neutrophil function is a potential concern. However, given that targeting G-CSF did not cause neutropenia and may actually resolve some underlying immunodeficiency, this is mitigated to some extent.

G-CSF has various hematological effects including the activation of endothelial cells[29], modulation of the leukocyte adhesion molecule expression[30,31], angiogenesis[20] and chemokine production[30]. Additionally, G-CSF also enhances neutrophil and macrophage phagocytosis[32,33] and prolongs neutrophil survival[34]. In agreement with this, we found enhanced myelopoiesis in the BM and also in extramedullary organs like the spleen. Therefore, in APLAID, overproduction of G-CSF might act both locally by mediating trafficking of myeloid cells through the endothelium, as well as promoting local cellular survival within inflamed sites and systemically via increased proliferation of hematopoietic progenitor cells in the BM and spleen and the release of mature neutrophils into the blood. Interestingly, while G-CSF levels in tissue lysates were undetectable, they remained elevated in plasma samples of *Plcg2*[S707Y/+]IL-6[−/−], *Plcg2*[S707Y/+]caspase-1[−/−] and *Plcg2*[S707Y/+]TNF[−/−] mice, highlighting the superior role of systemic effects over its contribution to the local milieu in our APLAID model. Furthermore, these data point toward a close interplay between G-CSF and each of these proinflammatory cytokine pathways and its multistep regulation of inflammasome and NF-kB activity, which is likely to be complex and multifactorial.

Nonetheless, the results from chimeras, where G-CSF was deleted from the non-hematopoietic compartment clearly demonstrate that this is where that cytokine is produced, in response to *PLCG2* mutant BM. Therefore, there should be a secreted factor, metabolite, or other cellular interaction, whereby the mutant *PLCG2* hematopoietic cell signals to the G-CSF-producing non-hematopoietic cell (Extended Data Fig. 8b).

PLCγ2 has an established role downstream of the B cell receptor[11,35] and other cell surface receptors[36–38], driving the conversion of $PIP_2$ to DAG, which can activate PKC and subsequent transcriptional responses[39,40]. However, the shared biological pathway between PLCγ2

**Fig. 6 | Bone marrow transplantation normalizes G-CSF levels and rescues phenotype in *Plcg2*[S707Y/+] mice. a**, Schematic of BM chimera generation (green). *Plcg2*[+/+] (blue) is the donor for lethally irradiated *Plcg2*[S707Y/+] recipients (yellow). Single-cell suspension of BM cells ($1 \times 10^6$ per ml) were transplanted by i.v. injection into recipient animals 3 h after irradiation. **b**, APLAID skin score demonstrates a rescued autoinflammatory phenotype in BM chimeras ($n = 6$) receiving a healthy transplant compared to *Plcg2*[S707Y/+] ($n = 3$). The arrow indicates the exact time of irradiation and reconstitution at 26 d after birth. **c**, Growth curve exhibiting a steady weight gain in BM chimeras ($n = 6$ mice) compared to controls ($n = 3$). **d**, BM chimeras lacked enlarged spleens ($n = 3$ per group). **e**, Kaplan–Meier survival curve of BM chimeras ($n = 7$) showing prolonged survival rates in comparison to *Plcg2*[S707Y/+] mice ($n = 3$). **f**, ADVIA analyzer data of the blood revealed reduced neutrophil/monocyte and increased lymphocyte counts in *Plcg2*[S707Y/+] mice after BM transplantation from a healthy donor ($n = 3$ mice per group). **g**–**i**, FACS analysis exhibited decreased numbers

of myeloid cells, particularly of neutrophils (Ly6G[+]CD11b[+]) in skin tissue ($n = 3$ mice per group). **j**, Quantification of plasma G-CSF measured by ELISA revealed normalized levels (*Plcg2*[S707Y/+], $n = 3$; BM chimera, $n = 6$). **k**, Anatomic pathology was restored in BM chimeras with sparse immune cell infiltrates in paws, ears, tails, lung, gut and spleen. One representative IHC section (from three independent experiments) of MPO[+] and F4/80[+] stains is shown. BM chimeras were 3 weeks old at the time of irradiation and reconstitution. Error bars represent the mean ± s.e.m. Statistical significance for skin score was determined by a two-way ANOVA. Statistical significance for longitudinal weight data was determined by a paired two-sided *t*-test. Spleen weights between two groups were determined by an unpaired two-sided Student's *t*-test. Statistical significance for the survival curve was determined by a Mantel–Cox test. Statistical significance for G-CSF levels and cell numbers between two groups were determined by an unpaired two-sided Student's *t*-test.

and G-CSF remains unknown. Further light has been shed on this issue by in vivo studies demonstrating that PKC activation can trigger a mouse model of inflammatory skin disease due to G-CSF[41]. Furthermore, PLCγ2 expression in neutrophils was found to be required for the development of arthritis in the K/BxN mouse model[42], which is also known to be G-CSF dependent[43]. Interestingly, the cell-permeable DAG

analog 1-oleoyl-2-acetyl-*sn*-glycerol has been observed to drive G-CSF transcription in human monocytes in vitro[44]. A soluble factor such as this could be a good candidate to signal from mutant *PLCG2* BM cells to trigger G-CSF transcription from a non-hematopoietic cell, as required in this model of APLAID (Extended Data Fig. 8b); however, this is highly speculative. Although further confirmation is required, we suggest this

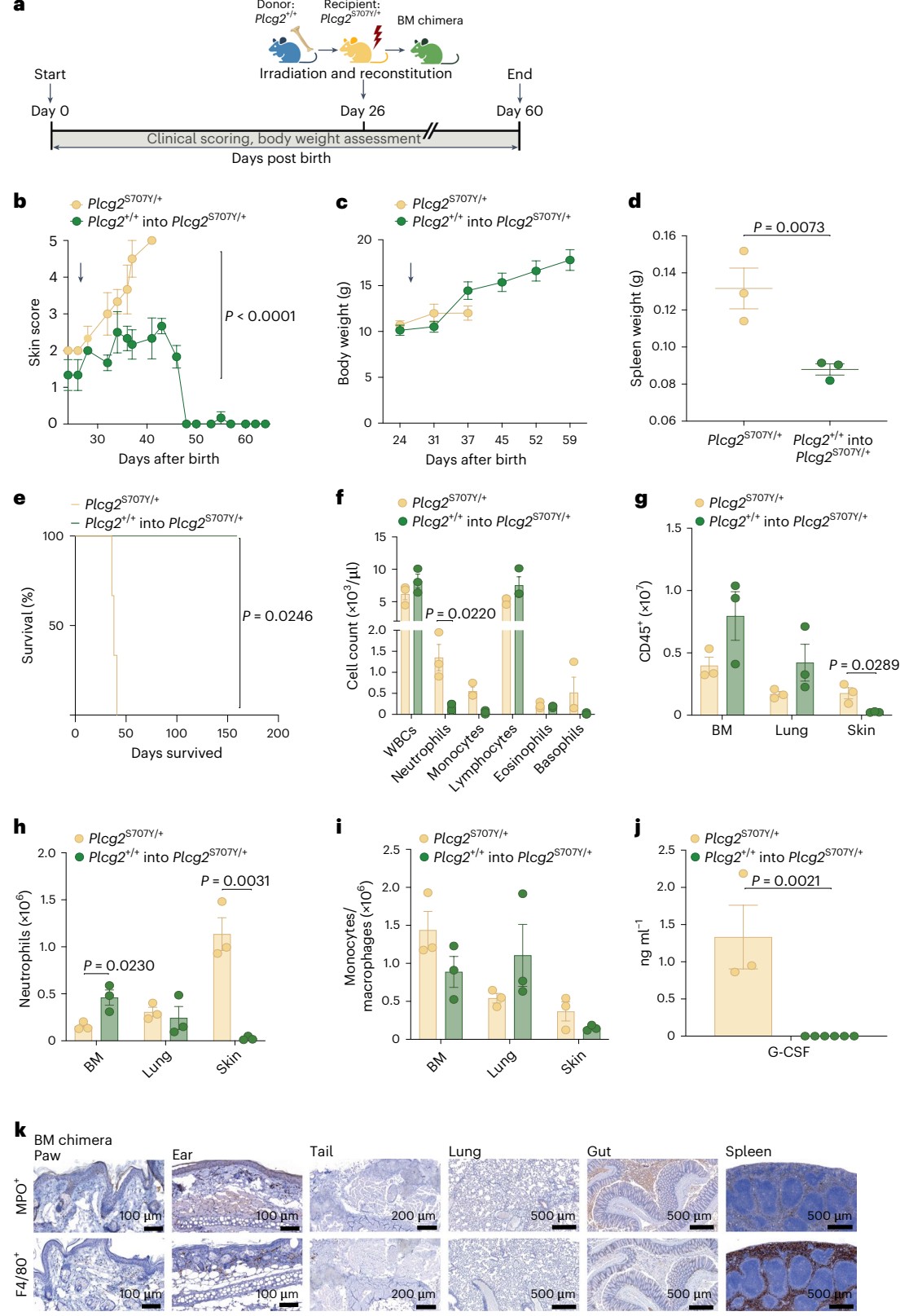

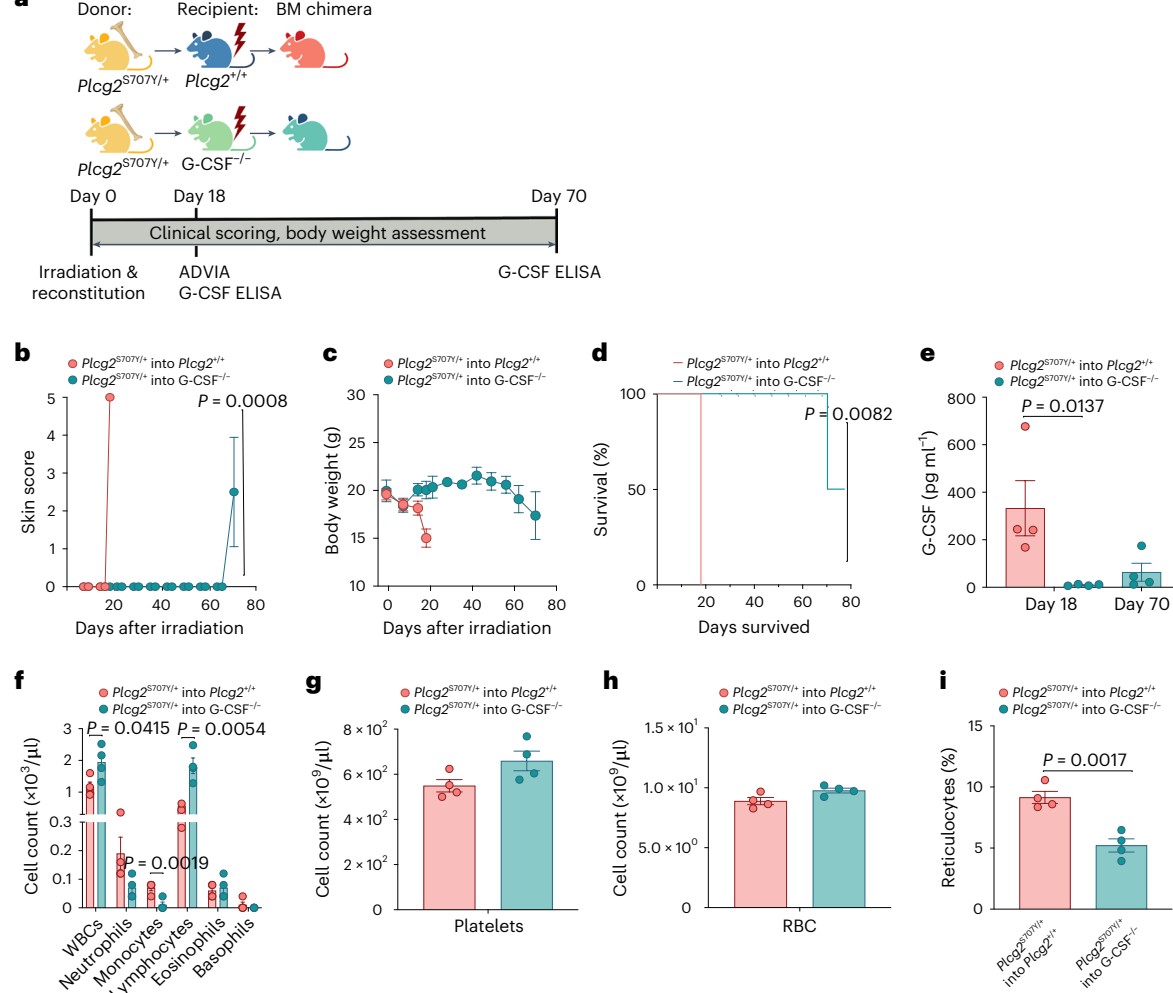

**Fig. 7 | Delayed onset of APLAID phenotype in G-CSF-deficient mice. a**, Schematic of BM chimera generation. *Plcg2*[S707Y/+] (yellow) is the donor for lethally irradiated WT (blue) or G-CSF[−/−] recipients (green). Single-cell suspension of BM cells ($1 \times 10^6$/ml) were transplanted by i.v. injection into recipient animals 3 h after irradiation. **b**, APLAID skin score revealed an autoinflammatory phenotype in WT mice receiving a *Plcg2*[S707Y/+] transplant at day 18, while G-CSF knockouts showed a beginning weight loss at day 70. **c**, Growth curve exhibiting dramatic weight loss in WT mice following irradiation and reconstitution with *Plcg2*[S707Y/+] BM cells. **d**, Kaplan−Meier survival curve showing prolonged survival rates of 52 d in G-CSF[−/−] mice receiving *Plcg2*[S707Y/+] BM cells. **e**, Quantification of plasma G-CSF measured by ELISA demonstrated elevated G-CSF levels in WT mice receiving *Plcg2*[S707Y/+] BM cells at day 18 and increasing G-CSF levels at the onset of clinical signs in G-CSF[−/−]

mice receiving *Plcg2*[S707Y/+] BM cells at day 70. **f−i**, ADVIA analyzer data of the blood at day 18 revealed increased neutrophil/monocyte and reduced lymphocyte counts in WT mice after BM transplantation from *Plcg2*[S707Y/+] mice, while thrombopoiesis and erythropoiesis remained unaltered except for reticulocytes. For all experiments, four mice aged 11 weeks per group were used. Error bars represent the mean ± s.d. in **c** and the mean ± s.e.m. for the remaining figures. Statistical significance for skin score was determined by a two-way ANOVA and by a paired two-sided *t*-test for longitudinal weight data. For survival data, statistical significance was determined by a Mantel−Cox test. G-CSF levels and cell numbers between two groups were determined by an unpaired two-sided Student's *t*-test. RBC, red blood cell; WBC, white blood cell.

mechanism could account for increased G-CSF production due to the gain-of-function *PLCG2* mutations causing APLAID.

In summary, by using a *Plcg2*[S707Y/+] mouse model, we identified a G-CSF-driven monogenic autoinflammatory disease. Treatment blocking G-CSF can reverse established autoinflammatory disease and restore B cells. Based on this preclinical model of APLAID, patients may benefit from allogeneic BM transplantation or neutralization of G-CSF.

## Online content

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

## Methods

### Generation of the *Plcg2*^S707Y/+ mouse model

The point mutation p.Ser707Tyr in *PLCG2* causes APLAID[1]. The presence of a Neomycin cassette prevents the expression of the knock-in point mutation. Only Flp-mediated excision enables the deletion of FRT-flanked Neomycin cassette, thereby generating the inflammatory phenotype. This deletion was performed in vivo, by breeding heterozygous PLCγ2 offspring harboring the recombined allele with C57BL/6 Flp deleter mice expressing ubiquitous Flp-recombinase[45]. The *Plcg2*^S707Y/+ mouse model was established by GSK.

### Mice

The following mouse strains were used: *Plcg2*^S707Y/+IL-6^−/− (ref. [46]), *Plcg2*^S707Y/+caspase-1^−/− (ref. [47]), *Plcg2*^S707Y/+TNF^−/− (ref. [48]), G-CSF^−/− (ref. [49]) and Ly5.1/J (ref. [50]). All genetically modified mouse strains in this study were on the C57BL/6J background, bred and maintained at the Walter and Eliza Hall Institute of Medical Research. Randomization was applied to mouse sex. Both male and female *Plcg2*^S707Y/+ mice at 26 weeks of age and their littermate controls were used for all experiments. Cohort sizes are described in each figure legend. All animal studies were ethically reviewed by and carried out in accordance with approval from the Walter and Eliza Hall Animal Committee (2017.029 and 2020.017).

### Genotyping

All genotyping of mouse strains was PCR based. Genomic tail DNA was obtained using DirectPCR Lysis Reagent (Viagen Biotech) per the manufacturer's instructions. PCR products were run on 1.5–2% agarose gels and visualized using ethidium bromide staining. Primers were purchased from Integrated DNA Technologies, and oligonucleotide details are provided in Supplementary Table 1.

### Clinical scoring

Clinical assessment for skin inflammation used the following scoring system: 0, no clinical signs of skin inflammation; 1, early signs of inflammation including flaky skin and scruffy fur; 2, skin lesions occur in one body region including paws or ears or tail and appear dry/not open; 3, skin lesions appear in two body regions (including paws and/or tail and/or ear) and present dry/not open; 4, skin lesions appear in three regions (including paws and/or tail and/or ear) and present dry/not open; 5, severe, open lesions are present in either of one above-mentioned body regions (paws, ear, tail), >20% loss of weight; scoring was performed in a blinded manner.

### Human samples

The study included sera of two affected APLAID individuals from two distinct families. Participant 1 has a heterozygous p.Ala708Pro mutation in *PLGC2* gene. In participant 2, the heterozygous mutation p.Leu845_Leu848del in *PLGC2* gene was detected. Written informed consent was obtained from all participants and healthy control volunteers. The Ethical Review Board of Hospital Clinic, Barcelona, Spain, approved the study (HCB/2019/0631). All investigations were performed in accordance with the ethical standards of the 1964 Declaration of Helsinki and its later amendments.

### Hematology

Automated cell counts were performed on mouse blood collected from the submandibular vein into Microtainer tubes containing EDTA (Sarstedt), using an ADVIA2120i hematological analyzer (Siemens).

### Flow cytometry

Cells were isolated after perfusion through the left cardiac ventricle with PBS. Skin tissue including paws, tail, ears and lungs were then incubated in digestion buffer (2 mg ml^−1 Collagenase IV (Worthington Biochemical), 1 mg ml^−1 Dispase (Worthington Biochemical) and 0.5 mg ml^−1 DNase I (Sigma) in PBS)) for 45 min at 37 °C, with agitation. Cells released during digestion were filtered through a 100-μm nylon mesh. Erythrocytes were lysed with red cell lysis buffer (156 mM NH$_4$Cl, 11.9 mM NaHCO$_3$, 0.097 mM EDTA). Cells were stained with viability dyes, incubated with FcR block, and stained with fluorochrome-conjugated antibodies (see Supplementary Table 1 for antibodies, clones, fluorochromes and manufacturers). Cells were washed and fixed with Cytofix/Cytoperm buffer (BD Biosciences) for 20 min on ice in the dark, then stored at 4 °C. Quantification of total cell numbers by flow cytometry was done using fluorescent beads (Beckman Coulter). Data were acquired on a BD Symphony A5 flow cytometer. Compensation and analysis of the flow cytometry data were performed using FlowJo software (Tree Star).

### BrdU assay

To assess BrdU incorporation into hematopoietic stem and progenitor cells, mice were i.p. injected with a single dose of 1 mg BrdU 2 h before organ collection. Cells were labeled with the indicated FACS antibodies, then fixed and stained with a BrdU antibody using the FITC-BrdU Flow kit (BD Biosciences) according to the manufacturer's protocol.

### Histology

Organs were collected in 10% neutral buffered formalin. Organ sections were prepared from paraffin blocks and stained with H&E. IHC was performed using the antibodies summarized in Supplementary Table 1.

### Computed tomography scan

For Micro-CT scans, mice were euthanized and scanned on a Bruker Skyskan 1276 Micro-CT. Images were analyzed using Fiji/BoneJ software.

### Multiplex magnetic bead-based immunoassays

Mouse plasma samples were collected and immediately stored at −20 °C. A total of 23 analytes were measured using the Bio-Plex Pro Mouse Cytokine 23-plex Assay. These include IL-1α, IL-1β, IL-2, IL-3, IL-4, IL-5, IL-6, IL-9, IL-10, IL-12 (p40), IL-12 (p70), IL-13, IL-17A, eotaxin, G-CSF, GM-CSF, IFN-γ, KC, MCP-1 (MCAF), MIP-1α/β, RANTES and TNF. The assay plate was pre-coated with beads labeled with fluorescent dye and was washed two times with Bio-Plex Wash Buffer on a Bio-Plex handheld magnetic washer. The plate was sealed with a sealing tape after samples and standards were added and incubated on a microtiter plate shaker for 30 min. After incubation, the plate was washed three times with wash buffer on a handheld magnetic washer. Biotinylated detection antibodies were incubated with the bead–antibody bound samples for 30 min and washed three times with wash buffer. Next, streptavidin conjugated to phycoerythrin was added and incubated for 10 min to allow formation of the final detection complex. Finally, the plate was washed three times, before the complexes were resuspended in assay buffer for acquisition on the Bio-Plex 200 array reader using the Bio-Plex Manager software. Five-parameter logistic function was used to generate the standard curve, in which a regression analysis was performed to report the fluorescence intensity and observed concentration of each analyte.

Human plasma samples were measured using the Milliplex Cytokine/Chemokine/Growth Factor Panel A-Immunology Multiplex Assay, according to manufacturer's protocol. Catalog numbers and manufacturers' details of both commercial kits are provided in Supplementary Table 1.

### ELISA

All ELISA experiments (G-CSF, M-CSF from R&D and immunoglobulins subtypes from Thermo Fisher Scientific) were carried out according to the manufacturer's instructions. Catalog numbers and manufacturers' details are given in Supplementary Table 1.

### Colony assay

Single-cell suspensions from BM and spleen were prepared in balanced salt solution (0.15 M NaCl, 4 mM KCl, 2 mM CaCl$_2$, 1 mM MgSO$_4$, 1 mM

KH$_2$PO$_4$, 0.8 mM K$_2$HPO$_4$ and 15 mM *N*-2-hydroxyethylpiperazine-*N'*-2-ethanesulfonic acid supplemented with 2% (vol/vol) bovine calf serum). Clonal analysis of BM cells ($2.5 \times 10^4$) and splenocytes ($5 \times 10^4$) was performed in 1 ml semisolid agar cultures of 0.3% agar in DMEM containing 20% newborn calf serum, stem cell factor (100 ng ml$^{-1}$; in-house), erythropoietin (2 U ml$^{-1}$; Janssen), IL-3 (10 ng ml$^{-1}$; in-house), G-CSF (103 U ml$^{-1}$; PeproTech), granulocyte–macrophage colony-stimulating factor (103 U ml$^{-1}$; in-house) and/or M-CSF (103 U ml$^{-1}$; in-house). Cultures were incubated at 37 °C for 7 d in a fully humidified atmosphere of 10% CO$_2$ in air, then fixed, dried onto glass slides and stained for acetylcholinesterase, Luxol fast blue, hematoxylin and the number and type of colonies were determined.

## G-CSF antibody treatment

Mice were injected i.p. with an anti-G-CSF rat IgG1 monoclonal antibody (clone MAB414, 50 µg per injection, R&D) to neutralize G-CSF or an IgG1 isotype control (clone MAB005, 50 µg per injection, R&D). Antibody injections were given three times a week starting on day 14 after birth for a total of four consecutive weeks.

## Bone marrow chimeras

BM was flushed from the femurs and filtered through a 100-µm filter. A total of $1 \times 10^6$ cells in 200 µl PBS were intravenously injected into lethally irradiated ($2 \times 550$ rads) mice.

## Cell sorting

FACS sorting of primary mouse cells was performed by staff of the Walter and Eliza Hall Institute of Medical Research flow cytometry core facility (Melbourne, Australia). Depending on the experimental requirements and instrument availability, the BD FACSAria W (lasers: 375 nm, 405 nm, 488 nm, 561 nm and 640 nm) and/or BD FACSAria Fusion (lasers: 405 nm, 488 nm, 561 nm and 640 nm; BD Biosciences) were used. Before sorting, cells were resuspended in 1× PBS/2% FCS and kept on ice. Neutrophils (CD45$^+$, CD11b$^+$, Ly6G$^+$, F4/80$^-$, CD115$^-$, CD64$^-$), macrophages (CD45$^+$, CD11b$^+$, F4/80$^+$, CD115$^+$, CD64$^+$, Ly6G$^-$) and stromal cells (endothelial cells: CD45$^-$, CD31$^+$; fibroblasts: CD45$^-$, CD31$^-$, gp38$^+$ PDGFRa$^+$; keratinocytes: CD45$^-$, CD31$^-$, PDGFRa$^-$, CD49f$^+$; and epithelial cells: CD45$^-$ CD31$^-$, PDGFRa$^-$, EPCAM$^+$) were directly sorted into 1.5-ml microcentrifuge tubes containing RLT lysis buffer (RNeasy Plus Micro Kit; QIAGEN) using a 70-µm or 100-µm nozzle as required.

## RNA isolation and quantitative real-time PCR

Total RNA was extracted from sorted cells using an RNeasy Plus Micro Kit (QIAGEN) and reverse-transcribed into complementary DNA using SuperScript III reverse transcription and oligo(dT) nucleotides (Thermo Fisher Scientific). Real-time qPCR was performed using SYBR Green/ROX qPCR Master Mix (Thermo Fisher Scientific) on a ViiA 7 PCR System (Thermo Fisher Scientific). Cytokine expression from qPCR was calculated as gene expression relative to *gapdh* as ΔCT. Primers for the genes assessed are shown in Supplementary Table 1.

## RNA-seq

The sequencing workflow was performed as previously described[51]. Briefly, RNA from sorted neutrophils and macrophages (purity 99–100%) was extracted with an RNeasy Plus Micro Kit (QIAGEN) according to the manufacturer's protocol. Extracted RNA quality and quantity were assessed using an Agilent 2200 TapeStation System (Agilent) with RNA ScreenTapes (Agilent). Next-generation sequencing libraries were created with 10 ng of RNA from samples with distinct 18S and 28S peaks and RNA integrity number values greater than 7, using an NEBNext Ultra II Directional RNA Library Prep Kit from Illumina (New England Biolabs) according to the manufacturer's protocol. RNA libraries were pooled and sent for single-end 75-bp sequencing at the Genomics Laboratory of The Walter and Eliza Hall Institute of Medical

Research on a NextSeq 500 next-generation sequencer (Illumina) to obtain ~20 million reads per sample.

## Bioinformatics

Paired-end 75-bp RNA-seq short reads were generated using a NextSeq 500 (Illumina). Between 16 and 64 million read pairs were generated for each sample, and reads were aligned to the *Mus musculus* genome (GRCm39/mm39) using Rsubread[52]. The number of read pairs overlapping mouse genes was summarized using featureCounts and Gencode (vM27) annotation. Lowly expressed genes were filtered out using edgeR's filterByExpr function[53]. Sex-linked genes and genes without current annotation were also filtered. Differential expression analysis was undertaken using the edgeR and limma[54] software packages. Library sizes were normalized using the quantile normalization method[55] using limma's normalizeBetweenArrays function. Sample relative quality weights were estimated using arrayWeights[56], and differential expression was evaluated using limma trend[57] with robust empirical Bayes estimation of the variances[58]. The effect caused by age differences of the mice was adjusted and correlations between repeated measurements from the same mouse were estimated using the duplicateCorrelation method[59]. The false discovery rate was controlled below 0.1 using the method of Benjamini and Hochberg. Overrepresentation of Gene Ontology terms and KEGG pathways for the differentially expressed genes were identified using limma's goana and kegga functions. Heat maps were drawn using the pheatmap function. Macrophage samples and neutrophil samples were analyzed separately. Enrichment of gene sets[21] was tested using the roast method[60] and illustrated using barcode plots drawn by limma's barcodeplot function.

## Statistics and reproducibility

No statistical methods were used to predetermine sample sizes, but our sample sizes are similar to those reported in previous publications[61]. Data distribution was assumed to be normal, but this was not formally tested. Differences in clinical score between the groups were assessed using two-way ANOVA with Bonferroni post-test correction for repeated measurements. For other experiments, differences between groups were determined using unpaired or paired two-sided Student's *t*-tests as indicated. The statistical test for the survival curve was determined by a Mantel–Cox test. No data points were excluded from the analyses.

All analyses were performed using GraphPad Prism v9.4.1. Each dot represents one mouse or one individual. Data are shown as means ± s.e.m. unless stated otherwise. Confidence intervals were set to 95%. Differences were statistically significant when *P* values were less than 0.05.

## Reporting summary

Further information on research design is available in the Nature Portfolio Reporting Summary linked to this article.

## Data availability

RNA-seq data have been deposited in the Gene Expression Omnibus under accession https://www.ncbi.nlm.nih.gov/geo/query/acc.cgi?acc=GSE211109. Biological materials that are not commercially available can be requested from the authors under a material transfer agreement. Source data are provided with this paper.

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

## Acknowledgements

We are grateful to all members of the laboratory of S.L.M. We thank the WEHI core facilities of flow cytometry (S. Monard), histology (E. Tsui and E. Pan), genomics (S. Wilcox) and imaging (L. Whitehead) for excellent technical support. We thank S. Russo, E. Azzopardi and L. Wilkins for outstanding animal husbandry. This work was supported by grants from the Deutsche Forschungsgemeinschaft (SCHU 3126/3-1; to E.M.), the Spanish Ministry of Science, Innovation and Universities co-funded by the European Regional Development Fund/ Agencia Estatal de Investigación (PID2021-125106OB-C31; to J.I.A.), the Australian National Health and Medical Research Council (1154325 and 1113577 to I.P.W.; 2003159 and 2003756 to S.L.M.), fellowships from the Victorian Endowment for Science Knowledge and Innovation (to S.L.M.), the HHMI-Wellcome International Research Scholarship (to S.L.M.) and the Sylvia and Charles Viertel Foundation (to S.L.M.). Figs. 6a and 7a and Extended Data Figs. 4a, 7b and 8b were created with BioRender.com.

## Author contributions

E.M. and S.L.M. designed the study. E.M., K.K., P.J.B., J.T.J., M.H., I.P.W., C.L. and S.L.M. performed and supervised the mouse experiments. J.I.A. collected clinical data and analyzed participant samples. A.N. performed the colony assay. S.-L.N. and J.N. designed the mouse model. N.K., A.L.G. and W.A. performed bioinformatic analysis. E.M. and S.L.M. wrote the manuscript with input from all authors.

## Competing interests

S.L.M. is a Scientific Advisor for Odyssey Therapeutics and NRG Therapeutics. S.-L.N. and J.N. are employees of GSK. I.P.W. has acted as a Scientific Advisor for CSL and received funding for research on G-CSF antagonism in inflammatory diseases. All other authors declare no competing interests.

## Additional information

**Extended data** is available for this paper at https://doi.org/10.1038/ s41590-023-01473-6.

**Correspondence and requests for materials** should be addressed to Seth L. Masters.

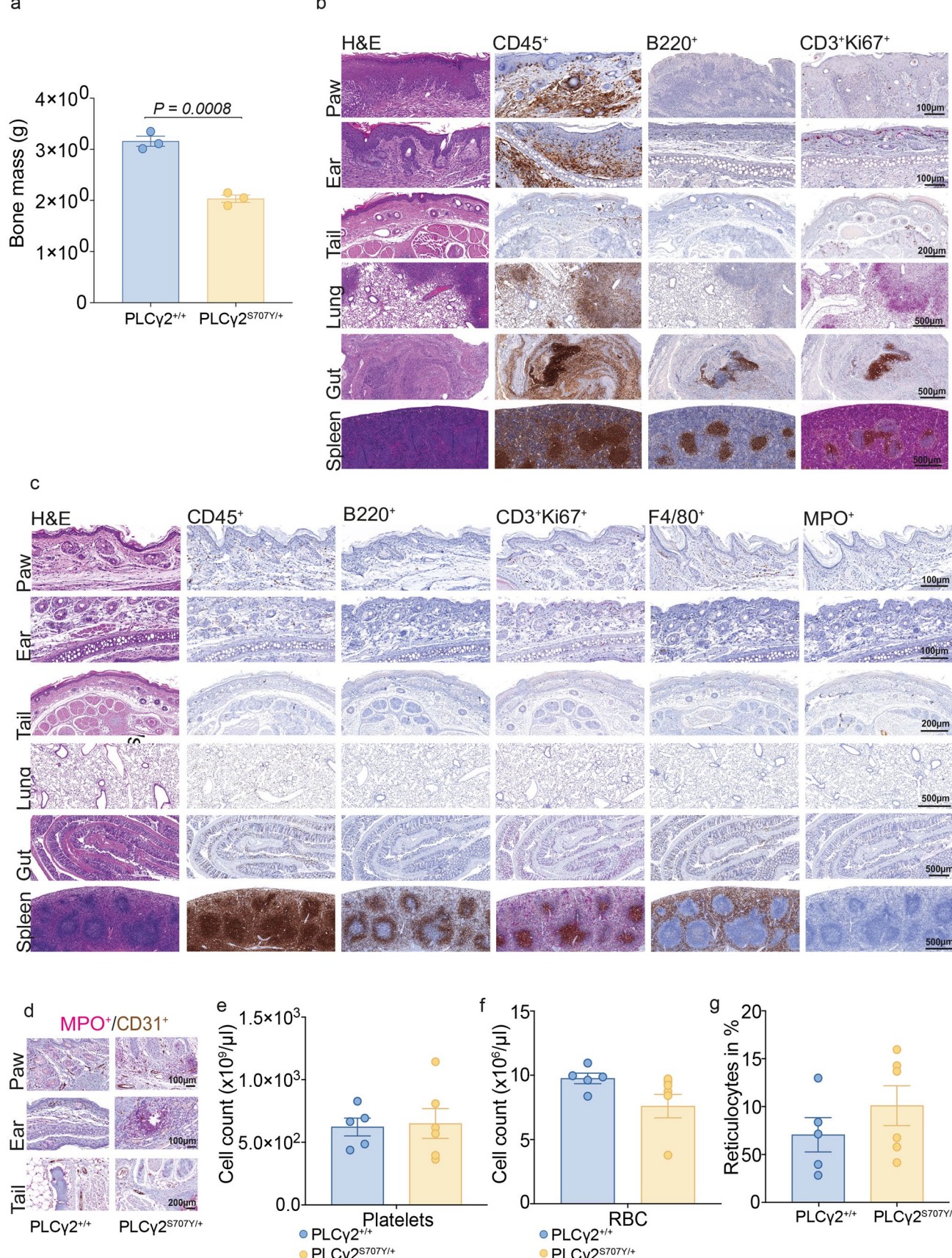

**Extended Data Fig. 1 | See next page for caption.**

**Extended Data Fig. 1 | Further validation of PLCγ2$^{S707Y/+}$ mice. a)** Total bone mass of 3-week-old, male PLCγ2$^{S707Y/+}$ mutants compared to littermate controls (n = 3 mice per group). **b)** Whilst increased numbers of CD45$^+$ across all organs are present, lymphoid immune cell infiltrates are sparse in paws, ears, tails, lung, gut, spleen. One representative IHC section (from three independent experiments; at 6 weeks of age) of CD45$^+$, CD3$^+$Ki67, B220$^+$ stains is shown. **c)** PLCγ2$^{+/+}$ littermate controls lack immune cells infiltrates. One representative IHC section (from three independent experiments; at 6 weeks of age) of H&E, CD45$^+$, MPO$^+$, F4/80$^+$, CD3$^+$Ki67, B220$^+$ is shown. **d)** Neutrophils (MPO$^+$ in magenta) reside predominantly in the tissue rather than in the blood endothelial vessels (CD31$^+$ with DAB) of 4-week-old PLCγ2$^{S707Y/+}$ mice as demonstrated by dual IHC (one representative from three independent experiments). **e-g)** ADVIA analyzer data of the blood reveal no difference with regards to platelets, red blood cells and reticulocytes in PLCγ2$^{S707Y/+}$ mutants compared to littermate controls (PLCγ2$^{+/+}$: n = 5; PLCγ2$^{S707Y/+}$: n = 6). Error bars represent mean ± SEM.

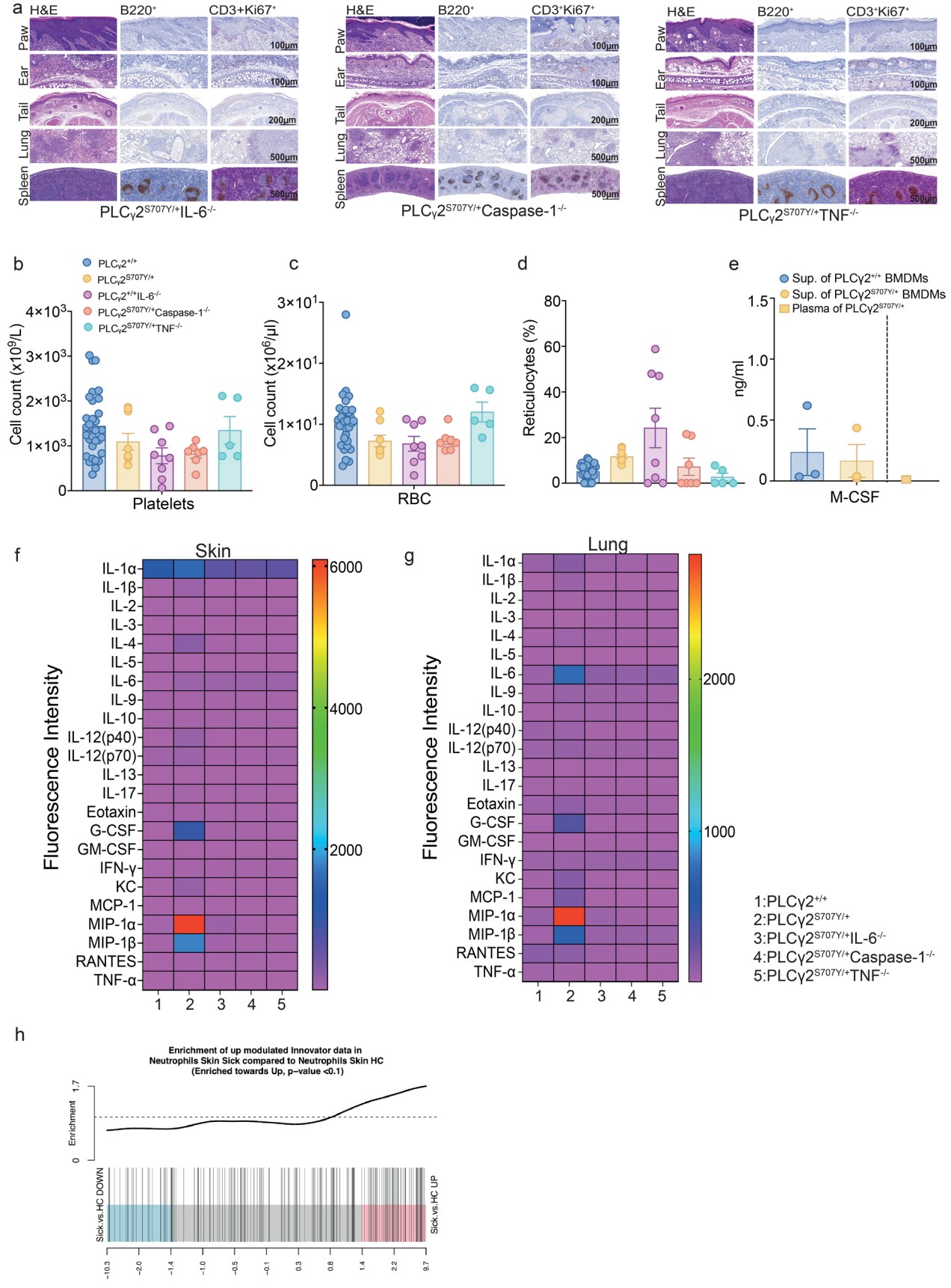

**Extended Data Fig. 2 | See next page for caption.**

**Extended Data Fig. 2 | Phenotype of PLCγ2$^{S707Y/+}$IL-6$^{-/-}$, PLCγ2$^{S707Y/+}$TNF$^{-/-}$ and PLCγ2$^{S707Y/+}$caspase-1$^{-/-}$. a)** While increased immune cell infiltrations are present across all organs, knock-out mice lack lymphoid immune cell infiltrates in skin and lung. One representative IHC section (from three independent experiments) H&E, CD3$^+$Ki67, B220$^+$ is shown. **b-d)** ADVIA analyzer data of the blood reveal no major differences with regards to platelets, red blood cells and reticulocytes in PLCg2$^{S707Y/+}$ knock-out strains (PLCγ2$^{+/+}$: n = 32; PLCγ2$^{S707Y/+}$: n = 9; PLCγ2$^{S707Y/+}$IL-6$^{-/-}$: n = 8; PLCγ2$^{S707Y/+}$caspase-1$^{-/-}$: n = 7; PLCγ2$^{S707Y/+}$TNF$^{-/-}$: n = 5; at 6 weeks of age). **e)** Quantification of M-CSF measured by ELISA were neither elevated in the supernatant of BMDMs nor in the plasma of PLCγ2$^{S707Y/+}$ mice (n = 3 mice per group; at 4 weeks of age). Error bars represent mean ± SEM.**f-g)** Cytokine assessment by a multiplex assay in tissue lysates from skin and lung reveal increased G-CSF levels followed by MIP-1α and MIP-1β only in PLCγ2$^{S707Y/+}$mice (n = 1) and not in PLCγ2$^{+/+}$(n = 1) PLCγ2$^{S707Y/+}$IL-6$^{-/-}$ (n = 3), PLCγ2$^{S707Y/+}$caspase-1$^{-/-}$ and PLCγ2$^{S707Y/+}$TNF$^{-/-}$ (n = 3) mice (n = 2) (at 5-6 weeks of age). Heatmap colours represent mean cytokine values. **h)** Enrichment of gene sets[21] are shown as barcode plots drawn by Limma's barcode plot function. Data are derived from 2 independent experiments with 4 (n = 5) and 6-week-old (n = 6) PLCγ2$^{S707Y/+}$mice and age-, number- and sex-matched littermate controls.

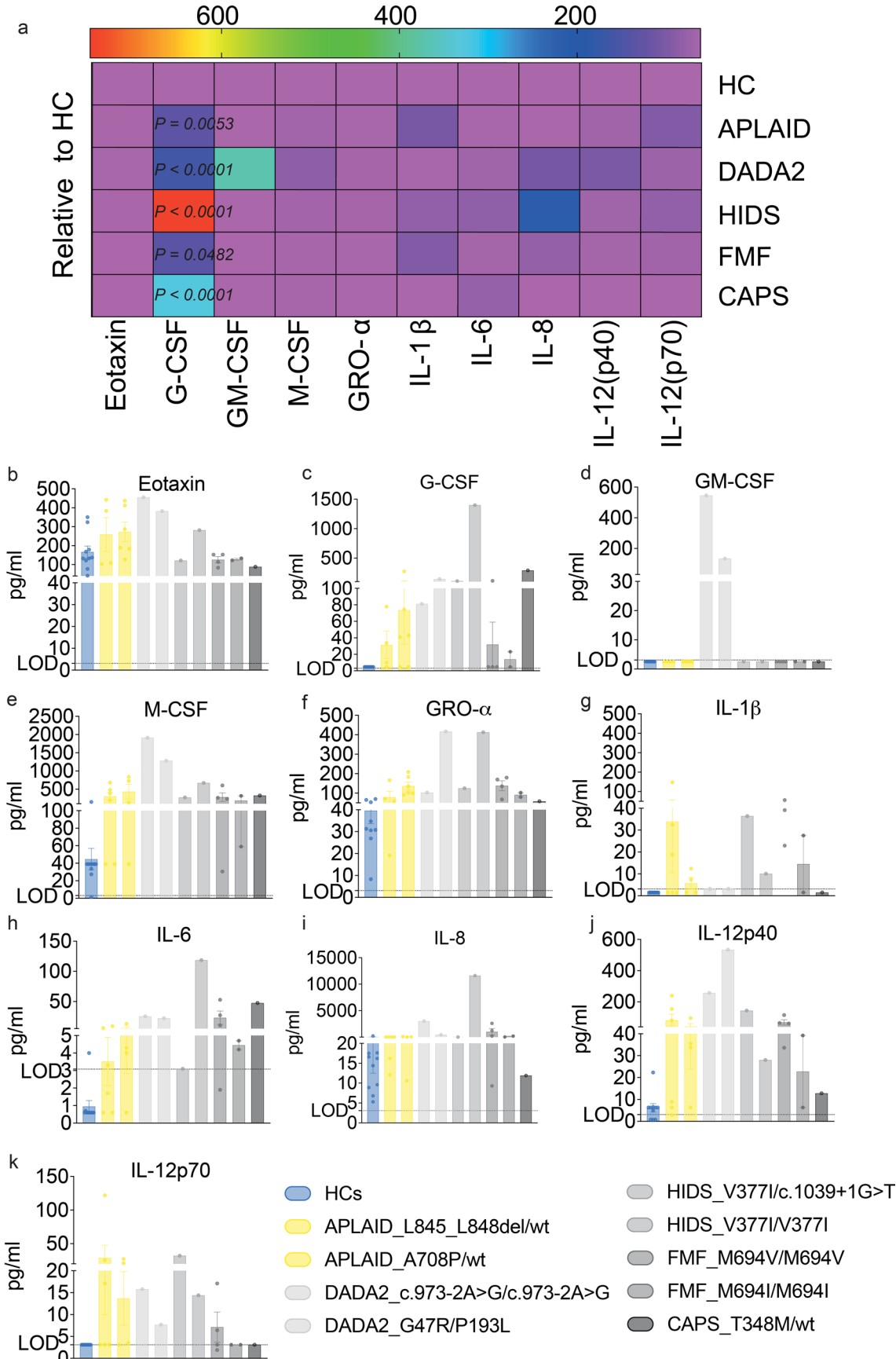

**Extended Data Fig. 3 | See next page for caption.**

**Extended Data Fig. 3 | G-CSF is not specific for APLAID.** Cytokine assessment by a multiplex assay in other autoinflammatory diseases including cryopyrin-associated periodic syndromes (CAPS) (n = 1/female/15-year-old), familial mediterranean fever (FMF) (n = 6/3females/3males/10-,45-,19-,19-.49-years-old), hyper-IgD with periodic fever syndrome (HIDS; n = 2/1female/1male/both 6-year-old), deficiency of adenosin deaminase-2 (DADA2) (n = 2/2 males/15 and 18-years-old) showed increased G-CSF levels relative to age- and sex-matched healthy control levels (n = 10). **a)** Colour mapping of cytokine heatmap represents mean values relative to control in pg/ml. **b-k)** Individual cytokine levels of all HC and patients with various autoinflammatory diseases are shown. The limit of detection (LOD) for each cytokine is indicated by a dotted line in each graph. Error bars represent mean ± SEM. Cytokine levels between two groups were determined by an unpaired two-sided Student's *t*-test.

a Experimental layout

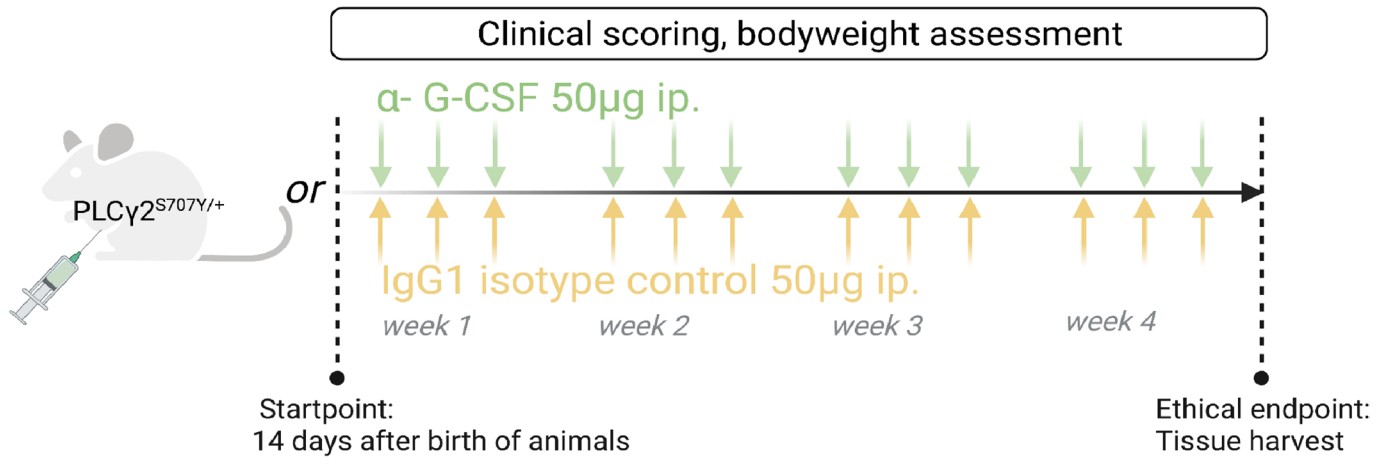

**Extended Data Fig. 4 | G-CSF blockade does not impact lympho-, erythro-and thrombopoiesis. a)** Anti-G-CSF treatment scheme in PLCγ2^S707Y/+ mice. **b)** Lymphocyte numbers also replenished in BM but not in spleen after anti-G-CSF treatment. **c-d)** ADVIA analyzer data of the blood reveal a decrease with regards to platelets and reticulocytes in PLCγ2^S707Y/+ mice following anti-G-CSF treatment whilst red blood cells remain unaltered (n = 3 mice per genotype). All mice were 2 weeks old at the start of the experiment. Error bars represent mean ± SEM.

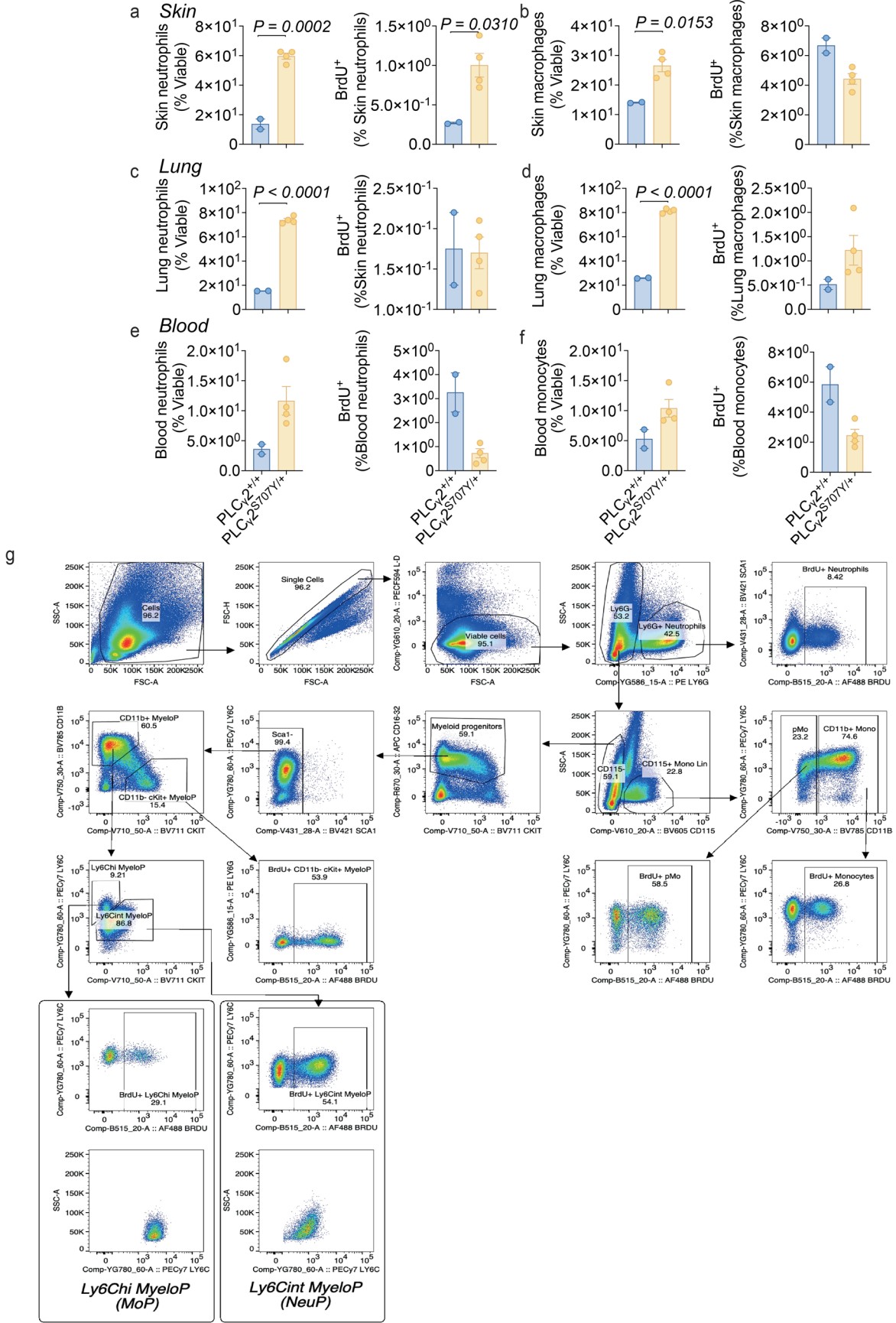

**Extended Data Fig. 5 | Skin, lung, and blood are not the primary sites of myelopoiesis in APLAID. a-f)** The myelopoiesis of progenitor and mature neutrophils and monocytes is not taking place in skin, lung and blood (PLCγ2[+/+]: n = 2; PLCγ2[S707Y/+]: n = 4). **g)** Representative FACS plots showing gating strategy used to identify BrdU[+] neutrophils and monocytes during different maturation stages in BM of a PLCγ2[S707Y/+] mouse. All mice were 4-5 weeks of age. Error bars represent mean ± SEM. Two to four mice were used per genotype. Cell numbers between two groups were determined by an unpaired two-sided Student's t-test.

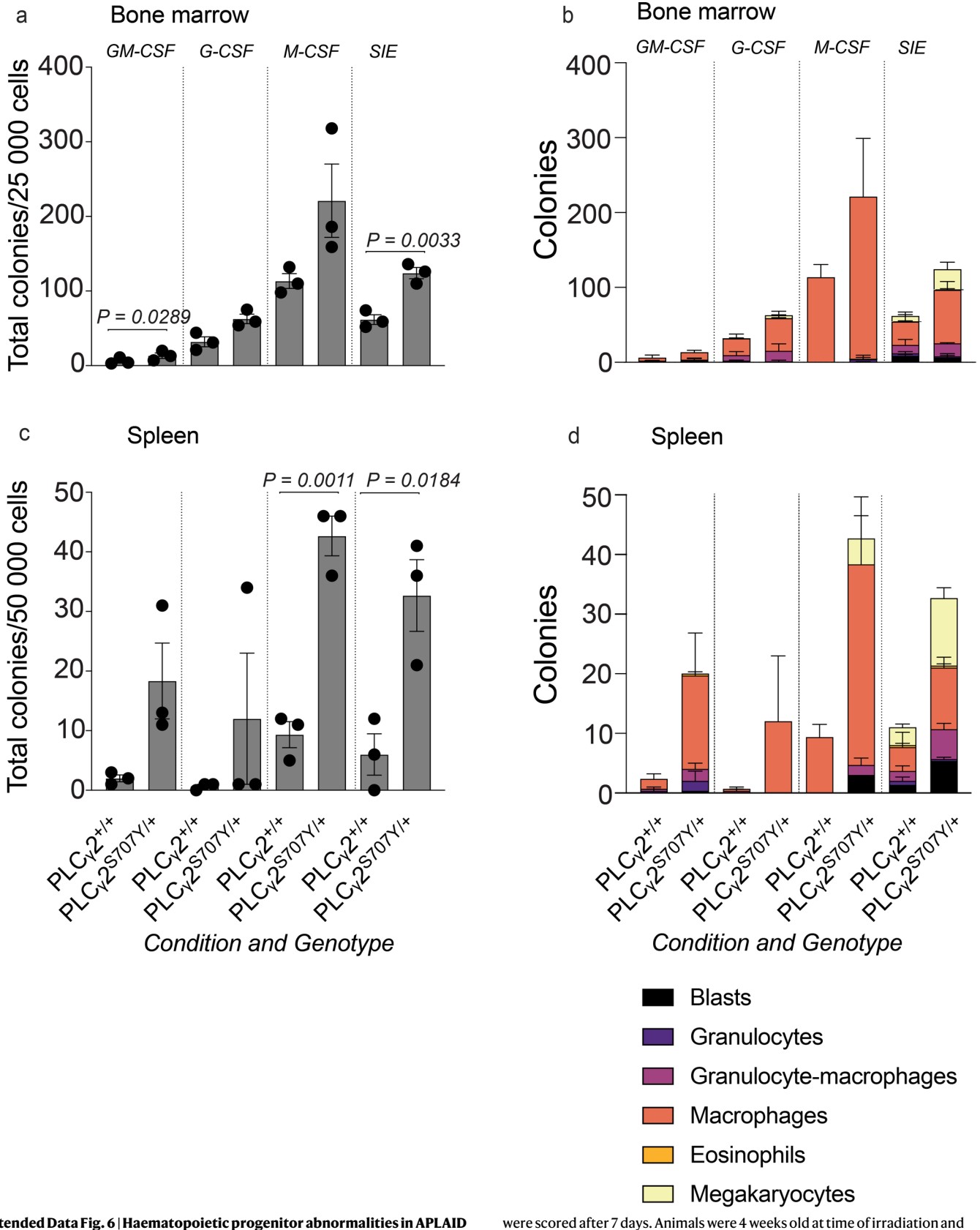

**Extended Data Fig. 6 | Haematopoietic progenitor abnormalities in APLAID indicated by a colony assays.** Type and numbers of colonies from 25,000 unfractionated BM cells and 50,000 splenocytes cultured in G-SCF (103 U/mL), GM-CSF (103 U/mL), SIE [SCF, (100 ng/mL), IL-3 (10 ng/mL), EPO (2 U/mL)] were scored after 7 days. Animals were 4 weeks old at time of irradiation and reconstitution. Error bars represent mean ± SEM. Statistical significance for total colonies was determined by an unpaired two-sided Student's $t$-test between groups (n = 3 mice per genotype).

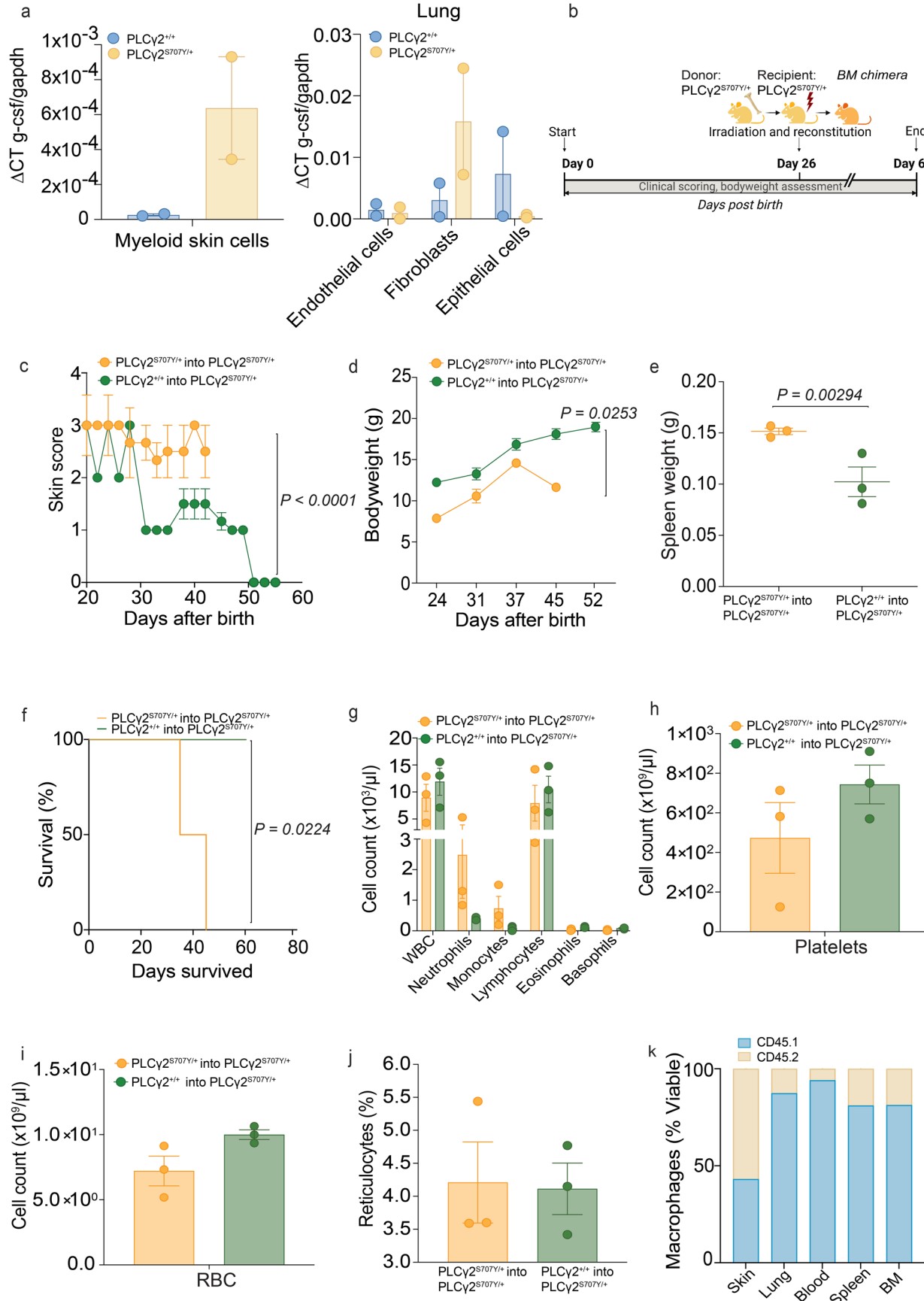

**Extended Data Fig. 7 | See next page for caption.**

**Extended Data Fig. 7 | The radioresistant compartment is dispensable for autoinflammation in APLAID. a)** RT-qPCR in FACS sorted immune cells from skin (left) and non-immune cells (right) from lung tissue in PLCγ2$^{S707Y/+}$ compared to PLCγ2$^{+/+}$ controls reveals elevated G-CSF transcripts in both skin myeloid cells (monocytes, macrophages, Langerhans Cells) and fibroblasts. Two independent experiments from 4 different mice per genotype are shown. Animals were 4–6 weeks old. **b)** Scheme of BM chimera generation (in dark yellow). PLCγ2$^{S707/+}$ (in yellow) are the donor for lethally irradiated PLCγ2$^{S707Y/+}$ recipients (yellow). Single cell suspension of BM cells (1 × 10$^6$/mL) are transplanted by i.v. injection into recipient animals 3 hrs after irradiation. **c)** Severity of skin inflammation is determined by the APLAID skin score on a 0–5 scale post weaning (n = 3 mice per group). **d)** A growth curve exhibits a continuous weight loss in PLCγ2$^{S707Y/+}$ into PLCγ2$^{S707Y/+}$ chimeras (n = 3 mice per group). **e)** Splenomegaly persists in the PLCγ2$^{S707Y/+}$ into PLCγ2$^{S707Y/+}$ chimera (n = 3 mice per group). **f)** Kaplan-Meier analysis demonstrate decreased survival rates in PLCγ2$^{S707Y/+}$ into PLCγ2$^{S707Y/+}$ chimera (n = 3 mice per group). **g-j)** ADVIA analyzer data of the blood reveal increased neutrophil and decreased lymphocyte counts in PLCγ2$^{S707Y/+}$ receiving BM from PLCγ2$^{S707Y/+}$ mice, whilst thrombo- and erythropoiesis remain unaltered (n = 3 mice per group). **k)** Proportion between circulating CD45.1$^+$ immune cells from the WT donor (6–8 weeks of age) and CD45.2$^+$ radioresistant immune cells from the recipient (4 weeks of age) 7 weeks post-transplantation (n = 3 mice per group). PLCγ2$^{S707Y/+}$ mice were 4-week-old. Error bars represent mean ± SEM. Statistical significance for skin score was determined by a two-way ANOVA. Statistical significance for longitudinal weight data was determined by a paired two-sided *t*-test. Spleen weights between two groups were determined by an unpaired two-sided Student's *t*-test. Statistical significance for the survival curve was determined by a Mantel-Cox test. G-CSF levels and cell numbers between two groups were assessed by an unpaired two-sided Student's *t*-test.

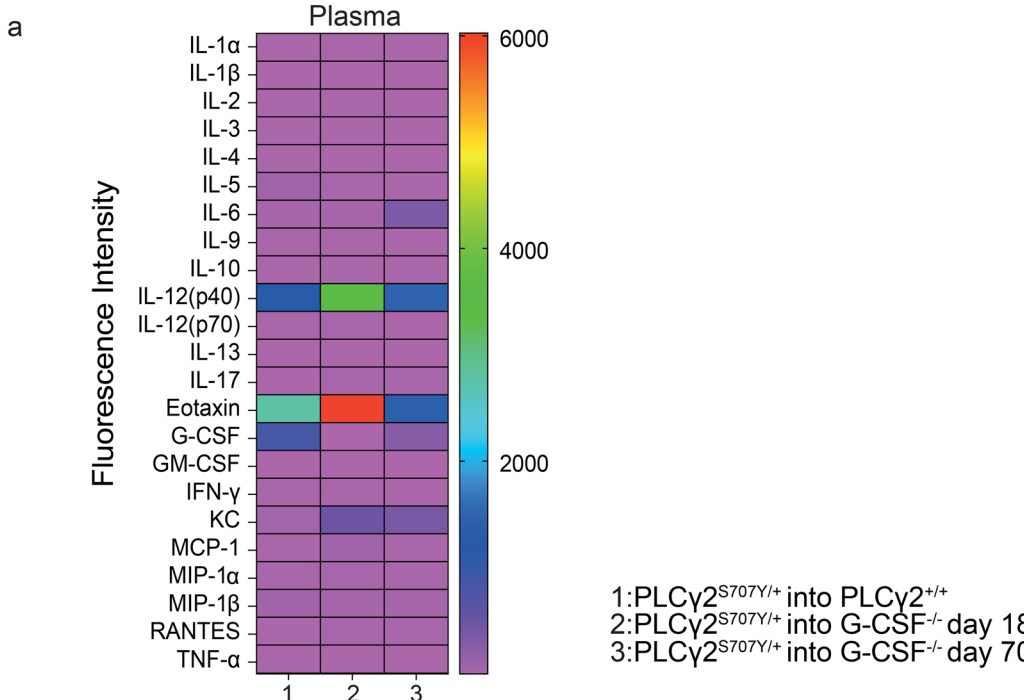

1:PLCγ2^S707Y/+ into PLCγ2^+/+
2:PLCγ2^S707Y/+ into G-CSF^-/- day 18
3:PLCγ2^S707Y/+ into G-CSF^-/- day 70

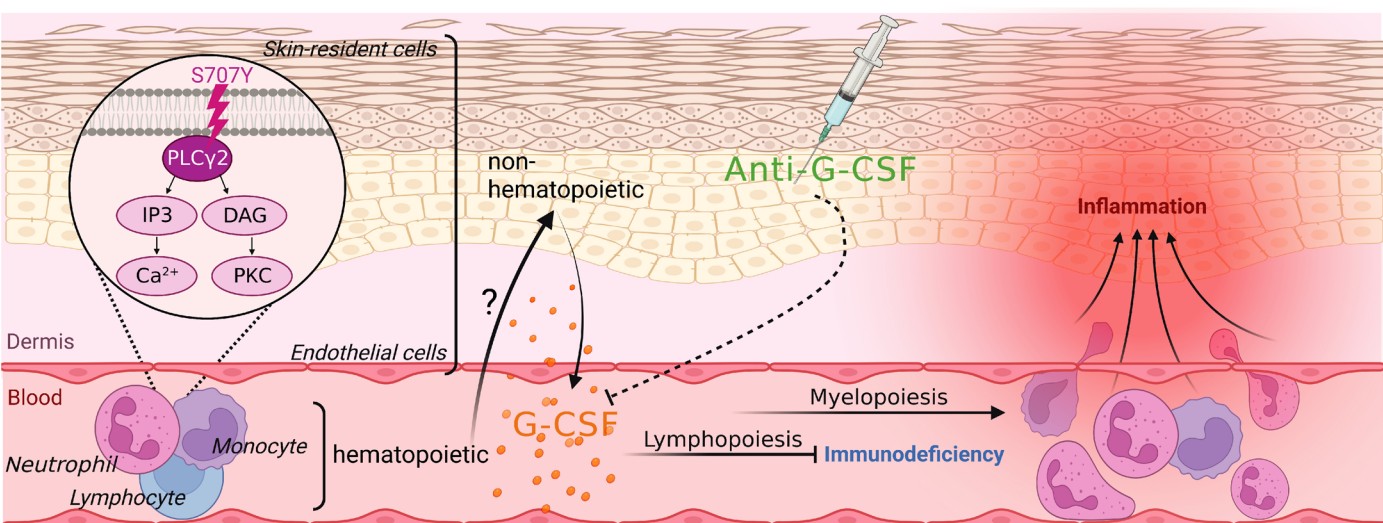

**Extended Data Fig. 8 | Proposed model of APLAID. a)** Cytokine assessment by a multiplex assay in plasma samples reveal increasing G-CSF levels following irradiation and reconstitution in PLCγ2^+/+ compared to G-CSF^−/− chimera (both groups at 11 weeks of age) receiving BM from PLCg2^S707Y/+ mice (n = 3 mice per group). **b)** Schematic illustration shows crosstalk between haematopoietic and non-haematopoietic cells in APLAID. Adapted from 'Basophil-Mediated Skin Inflammation', by BioRender.com (2023). Retrieved from https://app.biorender.com/biorender-templates.

**Extended Data Table 1 | Phenotypes/genotypes of published individuals with APLAID are summarized**

| | Zhou et al. [1] | Neves et al., 2018[2] | Novice et al., 2019[3] | Morán-Villaseñor et al., 2019[4] | Martín-Nalda et al., 2020[5] | Park et al., 2022[6] |
|---|---|---|---|---|---|---|
| Number of patients reported | 2 | 1 | 3 | 1 | 2 | 1 |
| Disease onset | Infancy | Birth | Infancy | Birth | Birth | Birth |
| Cutaneous lesions | Epidermolysis bullosa–like eruption, erythematous plaques, vesiculopustular lesions | Vesiculopustular rash, cutaneous granulomas, blistering skin rash, cutis laxa | Vesiculobullous, urticarial and nonspecific erythematous, polymorphic eruptions | Erythematous pustules, yellow-pink papules, pseudovesicles, acral hemorrhagic blisters, cutaneous granulomas | Erythematous plaques, Maculo-papular eruption, urticarial-like lesions, vesiculopustular and ulcerative lesions, ulcerative granulomata, hyperpigmentation cutis laxa | Blistery skin lesions |
| Eye inflammation | Corneal small blisters, corneal erosions, corneal ulcerations, intraocular hypertension cataracts | Recurrent eye inflammation, posterior uveitis, ocular hypertension | Episcleristis | Non-purulent conjunctival erythema | Bilateral corneal erosions, corneal limbitis, corneal nodules, haze bilateral conjunctivitis, bilateral episcleritis | - |
| Lung involvement | Interstitial pneumonitis | Interstitial pneumonitis | Chronic obstructive pulmonary disease | - | Bronchiectasis, recurrent episodes of hemoptysis | Asthma |
| Joint involvement | Arthralgias | | Arthritis | - | - | - |
| Gastrointestinal involvement | Enterocolitis, recurrent abdominal pain, bloody diarrhea, ulcerative colitis | Bloody diarrhea, early-onset inflammatory bowel disease | Bloody diarrhea | Recurrent episodes of diarrhea | - | - |
| Infections | Recurrent sinopulmonary infections, cellulitis | Recurrent infections (pneumonia, ear, sinus) | Recurrent viral and bacterial respiratory infections, recurrent urinary tract infections | Recurrent upper respiratory infections | Recurrent sinopulmonary infections, herpetic stomatitis, recurrent bacterial and fungal skin infections | Recurrent skin infections |
| Immunodeficiency | Yes | Yes | Yes | Yes | Yes | Yes |
| T cells | Normal | Normal | Normal | Normal | Normal | Normal |
| B cells | Normal/Low | Low | Normal/low | Low | Very low/absent | Low |
| NK cells | Normal | Normal | Normal | Low | Normal | Normal |
| IgG | Normal | Low | Absent | Low | Very low | Normal |
| IgA | Low | Low | Absent | Low | Very low | Normal |
| IgM | Low | Low | Low/absent | Low | Very low | Normal |
| IgE | n.a. | n.a. | n.a. | n.a. | Very low | Normal |
| Autoantibodies | Negative | Negative | Negative | Negative | Negative | Negative |
| PLCγ2 genotype | p.S707Y/WT | p.L848P/WT | p.M1141L | p.L848P/WT | p.L845_L848del/WT, p.A708P/WT | p.S707P/WT |

**Extended Data Table 2 | Phenotypes of different knockout strains**

| Mouse strain | Onset | Skin inflammation | Lung involvement | Gastrointestinal manifestation | Spleen size | Runtiness | Decreased blood lymphocyte counts | Elevated blood neutrophil counts | Sudden death |
|---|---|---|---|---|---|---|---|---|---|
| PLCγ2$^{S707/+}$ | Birth | Paws, ears, tail | Yes | Yes | Enlarged | Yes | Yes | Yes | Rarely |
| PLCγ2$^{S707Y/+}$IL-6$^{-/-}$ | Birth | Paws | Yes | Yes | Enlarged | No | Yes | Yes | No |
| PLCγ2$^{S707Y/+}$caspase-1$^{-/-}$ | Birth | Paws | Yes | Yes | Decreased | Yes | Yes | Yes | Yes |
| PLCγ2$^{S707Y/+}$TNF$^{-/-}$ | Birth, Post-weaning | Paws | Yes | Yes | Enlarged | No | Yes | Yes | No |

# Reporting Summary

## Statistics

For all statistical analyses, confirm that the following items are present in the figure legend, table legend, main text, or Methods section.

| n/a | Confirmed | |
|---|---|---|
| ☐ | ☒ | The exact sample size (*n*) for each experimental group/condition, given as a discrete number and unit of measurement |
| ☐ | ☒ | A statement on whether measurements were taken from distinct samples or whether the same sample was measured repeatedly |
| ☐ | ☒ | The statistical test(s) used AND whether they are one- or two-sided<br>*Only common tests should be described solely by name; describe more complex techniques in the Methods section.* |
| ☐ | ☒ | A description of all covariates tested |
| ☐ | ☒ | A description of any assumptions or corrections, such as tests of normality and adjustment for multiple comparisons |
| ☐ | ☒ | A full description of the statistical parameters including central tendency (e.g. means) or other basic estimates (e.g. regression coefficient) AND variation (e.g. standard deviation) or associated estimates of uncertainty (e.g. confidence intervals) |
| ☒ | ☐ | For null hypothesis testing, the test statistic (e.g. *F*, *t*, *r*) with confidence intervals, effect sizes, degrees of freedom and *P* value noted<br>*Give P values as exact values whenever suitable.* |
| ☒ | ☐ | For Bayesian analysis, information on the choice of priors and Markov chain Monte Carlo settings |
| ☒ | ☐ | For hierarchical and complex designs, identification of the appropriate level for tests and full reporting of outcomes |
| ☒ | ☐ | Estimates of effect sizes (e.g. Cohen's *d*, Pearson's *r*), indicating how they were calculated |

*Our web collection on statistics for biologists contains articles on many of the points above.*

## Software and code

Policy information about availability of computer code

| | |
|---|---|
| Data collection | No software was used. |
| Data analysis | Paired-end 75bp RNA-seq short reads were generated using NextSeq500 (Illumina). Between 16 and 64 million read pairs were generated for each sample and reads were aligned to the Mus musculus genome (GRCm39/mm39) using Rsubread 40. The number of read pairs overlapping mouse genes was summarized using featureCounts and Gencode (vM27) annotation. Low expressed genes were filtered out using edgeR's filterByExpr function 41. Sex link genes and the genes without current annotation were also filtered. Differential expression (DE) analysis were undertaken using the edgeR and limma 42 software packages. Library sizes were normalized using the quantile normalization method 43 using limma's normalizeBetweenArrays function. Sample relative quality weights were estimated using arrayWeights 44 and differential expression was evaluated using limma trend 45 with robust empirical Bayes estimation of the variances 46. The effect caused by age differences of the mice was adjusted and correlations between repeated measurements from the same mouse were estimated using the duplicateCorrelation method 47. The false discovery rate (FDR) was controlled below 0.1 using the method of Benjamini and Hochberg. Over-representation of Gene Ontology (GO) terms and KEGG pathways for the differentially expressed genes were identified using limma's goana and kegga functions. Heatmaps were drawn using pheatmap function. Macrophages samples and neutrophils samples were analyzed separately. Enrichment of gene sets 16 was tested using the roast method 48 and illustrated using barcode plots drawn by limma's barcodeplot function.<br>16. Avila-Portillo, L.M. et al. Comparative Analysis of the Biosimilar and Innovative G-CSF Modulated Pathways on Umbilical Cord Blood-Derived Mononuclear Cells. Bioinform Biol Insights 14, 1177932220913307 (2020).<br>40. Liao, Y., Smyth, G.K. & Shi, W. The R package Rsubread is easier, faster, cheaper and better for alignment and quantification of RNA sequencing reads. Nucleic Acids Res 47, e47 (2019).<br>41. Chen, Y., Lun, A.T. & Smyth, G.K. From reads to genes to pathways: differential expression analysis of RNA-Seq experiments using Rsubread and the edgeR quasi-likelihood pipeline. F1000Res 5, 1438 (2016). |

42. Ritchie, M.E. et al. limma powers differential expression analyses for RNA-sequencing and microarray studies. Nucleic Acids Res 43, e47 (2015).
43. Smyth, G.K. & Speed, T. Normalization of cDNA microarray data. Methods 31, 265-273 (2003).
44. Ritchie, M.E. et al. Empirical array quality weights in the analysis of microarray data. BMC Bioinformatics 7, 261 (2006).
45. Law, C.W., Chen, Y., Shi, W. & Smyth, G.K. voom: Precision weights unlock linear model analysis tools for RNA-seq read counts. Genome Biol 15, R29 (2014).
46. Phipson, B., Lee, S., Majewski, I.J., Alexander, W.S. & Smyth, G.K. Robust Hyperparameter Estimation Protects against Hypervariable Genes and Improves Power to Detect Differential Expression. Ann Appl Stat 10, 946-963 (2016).
47. Smyth, G.K., Michaud, J. & Scott, H.S. Use of within-array replicate spots for assessing differential expression in microarray experiments. Bioinformatics 21, 2067-2075 (2005).
48. Wu, D. et al. ROAST: rotation gene set tests for complex microarray experiments. Bioinformatics 26, 2176-2182 (2010).

For manuscripts utilizing custom algorithms or software that are central to the research but not yet described in published literature, software must be made available to editors and reviewers. We strongly encourage code deposition in a community repository (e.g. GitHub). See the Nature Portfolio guidelines for submitting code & software for further information.

## Data

Policy information about availability of data

All manuscripts must include a data availability statement. This statement should provide the following information, where applicable:
- Accession codes, unique identifiers, or web links for publicly available datasets
- A description of any restrictions on data availability
- For clinical datasets or third party data, please ensure that the statement adheres to our policy

The accession number for the RNASeq reported in this paper is GEO accession no. GSE211109.

## Human research participants

Policy information about studies involving human research participants and Sex and Gender in Research.

| | |
|---|---|
| Reporting on sex and gender | Only two patients in the study. |
| Population characteristics | Only two patients in the study. |
| Recruitment | All patients who were available were recruited. |
| Ethics oversight | The Ethical Review Board of Hospital Clínic, Barcelona, Spain, approved the study (HCB/2019/0631). |

Note that full information on the approval of the study protocol must also be provided in the manuscript.

# Field-specific reporting

Please select the one below that is the best fit for your research. If you are not sure, read the appropriate sections before making your selection.

☒ Life sciences　　☐ Behavioural & social sciences　　☐ Ecological, evolutionary & environmental sciences

For a reference copy of the document with all sections, see nature.com/documents/nr-reporting-summary-flat.pdf

# Life sciences study design

All studies must disclose on these points even when the disclosure is negative.

| | |
|---|---|
| Sample size | Patient data utilized all samples that were available to us. For in vitro experiments using cell lines, no sample size calculations were performed. For the majority of experiments, three biologically independent repeats were performed based on the number of replicates used in other studies using similar methods. |
| Data exclusions | There were no data exclusions unless a technical error was detected and flagged during the experiment. |
| Replication | The patient study has not yet been replicated due to limitations of sample availability.<br>For most experiments independent replicates were performed. |
| Randomization | Mice were randomly assigned to receive drug or control treatment. |
| Blinding | Mice were scored in a blinded fashion. |

# Reporting for specific materials, systems and methods

We require information from authors about some types of materials, experimental systems and methods used in many studies. Here, indicate whether each material, system or method listed is relevant to your study. If you are not sure if a list item applies to your research, read the appropriate section before selecting a response.

## Materials & experimental systems

| n/a | Involved in the study |
|-----|----------------------|
| ☐ | ☒ Antibodies |
| ☒ | ☐ Eukaryotic cell lines |
| ☒ | ☐ Palaeontology and archaeology |
| ☐ | ☒ Animals and other organisms |
| ☒ | ☐ Clinical data |
| ☒ | ☐ Dual use research of concern |

## Methods

| n/a | Involved in the study |
|-----|----------------------|
| ☒ | ☐ ChIP-seq |
| ☐ | ☒ Flow cytometry |
| ☒ | ☐ MRI-based neuroimaging |

## Antibodies

| Antibodies used | Please see Table S1 |
|-----------------|---------------------|
| Validation | Please see Table S1 |

## Animals and other research organisms

Policy information about studies involving animals; ARRIVE guidelines recommended for reporting animal research, and Sex and Gender in Research

| Laboratory animals | C57BL/6 mice. |
|--------------------|---------------|
| Wild animals | Study did not involve wild animals. |
| Reporting on sex | Both sexes reported. |
| Field-collected samples | No field collected samples. |
| Ethics oversight | All animal studies were ethically reviewed by, and carried out in accordance with approval from, the Walter and Eliza Hall Animal Ethics Committee (2020.017). |

Note that full information on the approval of the study protocol must also be provided in the manuscript.

## Flow Cytometry

### Plots

Confirm that:

☒ The axis labels state the marker and fluorochrome used (e.g. CD4-FITC).

☒ The axis scales are clearly visible. Include numbers along axes only for bottom left plot of group (a 'group' is an analysis of identical markers).

☒ All plots are contour plots with outliers or pseudocolor plots.

☒ A numerical value for number of cells or percentage (with statistics) is provided.

### Methodology

| Sample preparation | Skin tissue including paws, tail, ears and lungs were incubated in digestion buffer (2 mg/mL Collagenase IV, 1 mg/mL Dispase, and 0.5 mg/mL DNase I in PBS) for 45min at 37°C, with agitation. Cells released during digestion were filtered through 100μm nylon mesh. Erythrocytes were lysed with red cell lysis buffer (156 mM NH4Cl, 11.9 mM NaHCO3, 0.097 mM EDTA). Cells were stained with viability dyes, incubated with FcR Block, and stained with fluorochrome-conjugated antibodies (see Table S1 for antibodies, clones, fluorochromes, and manufacturers). Cells were washed and fixed with BD Cytofix/Cytoperm buffer (BD Bioscience) for 20mins on ice in the dark, then stored at 4°C. Quantification of total cell numbers by flow cytometry were done using fluorescent beads (Beckman Coulter). |
|---|---|
| Instrument | Data were acquired on a BD Symphony A5 flow cytometer. |

| Software | FlowJo 10.5 software |
|---|---|
| Cell population abundance | Gating strategy included in supplementary information Figure S6. |
| Gating strategy | Gating strategy included in supplementary information Figure S6. |

☒ Tick this box to confirm that a figure exemplifying the gating strategy is provided in the Supplementary Information.

