## [Peer Review File · Nature Immunology]

Peer Review Information

Journal: Nature Immunology

Manuscript Title: G-CSF drives autoinflammation in APLAID

Corresponding author name(s): Dr. Seth Masters

Editorial Notes:

Redactions – unpublished data	Parts of this Peer Review File have been redacted as indicated to maintain the confidentiality of unpublished data.
Redactions – confidential patient information	Parts of this Peer Review File have been redacted as indicated to maintain patient confidentiality.
Redactions – published data	Parts of this Peer Review File have been redacted as indicated to remove third-party material.
Redactions – reviewer opt-out	Parts of this Peer Review File have been redacted as indicated as we could not obtain permission to publish the reports of reviewer no. XX .
Reviewer comments in marked-up manuscript	In their review of the [first/second/third/...] version of this manuscript, reviewer no. XX added their comments to the manuscript file. These comments, excluding minor textual revisions, have been copied into this Peer Review File.

Reviewer Comments & Decisions:

Decision Letter, initial version:
--

4th Oct 2022

Dear Dr Masters,

Your Article, "G-CSF drives autoinflammation in APLAID" has now been seen by 2 referees. You will see from their comments below that while they find your work of interest, some important points are raised. We are very interested in the possibility of publishing your study in Nature Immunology, but would like to consider your response to these concerns in the form of a revised manuscript before we make a final decision on publication.

We therefore invite you to revise your manuscript taking into account all reviewer and editor comments. Please highlight all changes in the manuscript text file in Microsoft Word format.

* If you have not done so already please begin to revise your manuscript so that it conforms to our Article format instructions at <http://www.nature.com/ni/authors/index.html>. Refer also to any guidelines provided in this letter.

* Please include a revised version of any required reporting checklist. It will be available to referees to aid in their evaluation of the manuscript goes back for peer review. They are available here:

Reporting summary:

When submitting the revised version of your manuscript, please pay close attention to our [href="https://www.nature.com/nature-portfolio/editorial-policies/image-integrity">Digital Image Integrity Guidelines. and to the following points below:](https://www.nature.com/nature-portfolio/editorial-policies/image-integrity)

[redacted]

We hope to receive your revised manuscript within two months. If you cannot send it within this time, please let us know. We will be happy to consider your revision so long as nothing similar has been accepted for publication at Nature Immunology or published elsewhere.

Nature Immunology is committed to improving transparency in authorship. As part of our efforts in this direction, we are now requesting that all authors identified as 'corresponding author' on published papers create and link their Open Researcher and Contributor Identifier (ORCID) with their account on the Manuscript Tracking System (MTS), prior to acceptance. ORCID helps the scientific community achieve unambiguous attribution of all scholarly contributions. You can create and link your ORCID from the home page of the MTS by clicking on 'Modify my Springer Nature account'. For more information please visit www.springernature.com/orcid.

Sincerely,

Nick Bernard, PhD
Senior Editor
Nature Immunology

Reviewers' Comments:

Reviewer #1:

Remarks to the Author:

In their clearly and logically written and presented manuscript, "G-CSF drives autoinflammation in APLAID," Mulazzani et al. take a methodical and comprehensive approach to study the disease mechanism of a rare and often difficult to treat disease APLAID. They generated a novel mouse model that allowed for delineating pathways, testing of novel therapies, and identification of a new category of autoinflammatory disease. They also study APLAID patient blood during various treatments to support their conclusions.

The strengths of this paper are

1. Comprehensive phenotypic analysis of a novel mouse model of disease
2. Several useful knockout breedings and translational therapeutical trials including anti cytokine therapy and bone marrow transplant
3. In depth FACS based myelopoiesis studies
4. Data and interpretation that come together for an understandable story

The primary weaknesses

1. There are a few obvious and straightforward experiments that would strengthen the manuscript including breeding to GCSF ko and IL-1R ko
2. I would like to see a final model figure demonstrating mechanism

Comments:

Introduction

Concise and comprehensive and appropriate for this audience
Good review of current state of knowledge of this disease

Methods

Adequate detail of techniques but a few issues

1. Did GSK make this mouse? If so I didn't see any conflict of interest statements
2. Breeding strategies were not detailed enough as these mice do not live long enough to breed
3. Authors should state that the species of the anti GSCF antibody – I believe it is rat so there are likely allo-reactivity issues involved after long-term administration
4. Irradiation dose seems high
5. They need to distinguish skin neutrophils from neutrophils in the microcirculation (ie. Blood neutrophils in skin sections).
6. Are lung neutrophils including BALF? Was the lung perfused prior to dissociation and FACS analysis?

Results

Clear presentation of data but there are some errors

Fig 1. .

Fig 1i,j,l,m – were lung counts significant? Fig 1P – I don't see the reduction in Igs. If anything IgG 2a is increased

Fig 2.

Fig 2 b shows significant differences between KI and KI on IL-6 but TNF does not show significance but this is not stated

Fig 2 f again I do not see the Ig differences

Fig 2 g- what is going on with IL-12p40 in mouse

Fig 3

Cytokine heat maps should show individual data points including the limit of detection.

Consider moving supp Fig 3 into manuscript- comparisons to different autoinflammatory diseases is novel.

What is going on with IL-12p70 and IL-1b in human

Fig 4

Fig 4f. Igs are decreased with treatment

Fig 5 and Fig 6 could be combined

Fig 7

Obvious experiment is to do the opposite BMT control – Mutant BM into WT

Fig 7e appears to be a mistake - legend is switched? Colors should match the rest of the figure

Supplementary Figures

Mostly appropriate placement of supplemental data

Supp Fig 6. The FACS data needs compensation adjustment - cells cannot be sitting on the axis. Also needs a different format to convey how many cells are in these plots. Current format is hard to

interpret.

Supp Fig 6. The gating proposes a population of progenitors without the use of a lineage gate. This is essential. The gating for cKIT relative to the 'non-myeloid' population appears arbitrary and inappropriately set. The same is true for the Sca1 gate - are the authors claiming this entire population is Sca1 positive? Clear definitions of the progenitors needed here, and should conform to the literature in this regard unless otherwise justified.

Discussion

Data supports conclusions

Fonts are messed up

This section is written in a very choppy/sloppy manner unlike the rest of the manuscript

References - Adequate

Reviewer #2:

Remarks to the Author:

The manuscript by Mulazzani et al. describes novel findings that substantially expand our understanding of development of APLAID – dominantly inherited, monogenic immune disorder. Furthermore, the data suggest new therapeutic options linked to the high levels of G-CSF as a critical mediator in this pathology.

The evidence is comprehensive and experimental approaches and results appear to be without major problems. Some additional data and extensive improvements of the manuscript text are needed to further support and clearly present the findings and, importantly, to outline their wider relevance.

Specific points of criticism:

(1) Background information presented in Introduction should be improved. Specific points are below.

- Only 2 clinical studies for APLAID are referenced (ref 1 and 2). Because the number of all reported APLAID cases is relatively small - all should be referenced. Adapting and updating Table 1 in reference 2, as a supplemental information, is needed for the understanding of APLAID by broader readership.

- Signalling context of PLC γ 2 is not well explained. Notably - BLNK is not a tyrosine kinase. Statements should also be more specific about the cell types and receptors and distinguish B cells from other types. References 9, 10 and 11 are outdated; there are more recent relevant reviews focused on PLC. Similarly, description of the 3D structural organization should be based on more recent direct studies of PLC γ 1 structure and models of PLC γ 2 structure (see AF-P16885-F1), not ref 12.

- Introduction to the previously proposed link to NLRP3 should better cover and explain data in refs 2 and 13.

- The authors should briefly mention information gained from studies described in refs 14 and 15, not just the limitations of this work.

(2) Figure 1 presents comprehensive analyses of PLCy2S707Y/+ mice. It is also stated that “the clinical phenotype of PLCy2S707Y/+ mice strongly recapitulates the human disease”. Some further clarification of aspects that may not be “strongly recapitulated” should be provided. For example, are the findings shown in Figure 1p expected, based on data from patients?

(3) Description of the data obtained from crosses resulting in PLCy2S707Y/+IL-6-/-, PLCy2S707Y/+caspase-1-/- and PLCy2S707Y/+TNF-/- mice, needs further clarification. Specifically, know features of the phenotypes of the KO mice crossed with PLCy2S707Y/+ mice should be included as a supplemental table.

(4) The author’s state: “Multiplex cytokine assessment of skin and lung lysates also revealed increased G-CSF and MIP-1 α/β in PLCy2S707Y/+mice, but not in PLCy2S707Y/+IL-6-/-, PLCy2S707Y/+TNF-/- and PLCy2S707Y/+caspase-1-/- mice (Supp.Fig.2f-g)”. Considering the importance of G-CSF highlighted in this study, some further explanation should be provided about the implication of this difference between PLCy2S707Y/+mice and the crosses.

(5) The main conclusion from the genetic approach rules out the key importance of IL-6, inflammasome or TNF in driving APLAID. Considering that the evidence for the role of G-CSF has been obtained using pharmacological tools, it would be helpful to have a similar information that complements the genetic studies; specifically, the effect on PLCy2S707Y/+ mice of relevant pharmacological agents used in the clinic that block IL-1, JAK1/2 or TNF. These should be addressed experimentally. Conversely, it would be informative to know the phenotype of G-CSF KO mice.

(6) The information related to 2 clinical cases in Figure 3 doesn’t provide a direct or conclusive support for the key role of G-CSF in APLAID. Nevertheless, this information is interesting but should be better explained and discussed, in particular, the implications of the high G-CSF not being unique for APLAID monogenic disease or even less prominent compared to other immune disorders. Related to this, it is not clear, based on a small number of cases for each disorder, what are the differences between the individuals with the same disorder compared to different disorders.

(7) Data shown in Figures 5 and 6 and the related text could be better linked under one subheading in the result section.

(8) Discussion should be improved and include/elaborate some of the points.

- The authors should better connect known and new observations into possible steps of the disease development, linking cellular and organ/tissue levels.

- The role of G-CSF in non-monogenic inflammatory diseases should be explained and any parallels with APLAID highlighted.

Author Rebuttal to Initial comments

Response to Reviewer Comments

We would like to thank the editor and the reviewers for their very constructive questions that have helped us to improve the quality of this manuscript. We are now implementing their comments and suggestions and wish to submit a revised version of the manuscript

for further consideration. Below, we provide our proposed revisions in a point-by-point response.

Reviewer #1:

Remarks to the Author:

In their clearly and logically written and presented manuscript, "G-CSF drives autoinflammation in APLAID," Mulazzani et al. take a methodical and comprehensive approach to study the disease mechanism of a rare and often difficult to treat disease APLAID. They generated a novel mouse model that allowed for delineating pathways, testing of novel therapies, and identification of a new category of autoinflammatory disease. They also study APLAID patient blood during various treatments to support their conclusions.

The strengths of this paper are

1. Comprehensive phenotypic analysis of a novel mouse model of disease
2. Several useful knockout breedings and translational therapeutical trials including anti cytokine therapy and bone marrow transplant
3. In depth FACS based myleopoiesis studies
4. Data and interpretation that come together for an understandable story

The primary weaknesses

1. There are a few obvious and straightforward experiments that would strengthen the manuscript including breeding to GCSF ko and IL-1R ko
2. I would like to see a final model figure demonstrating mechanism

Comments:

Introduction

Concise and comprehensive and appropriate for this audience
Good review of current state of knowledge of this disease

Methods

Adequate detail of techniques but a few issues

1. Did GSK make this mouse? If so I didn't see any conflict of interest statements.

The mouse model was indeed generated by GSK and that is now stated explicitly on page 18. Conflicts of interest with regards to GSK are now stated on p.25.

2. Breeding strategies were not detailed enough as these mice do not live long enough to breed

The breeding strategy is now expanded (p.18).

3. Authors should state that the species of the anti GSCF antibody – I believe it is rat so there are likely allo-reactivity issues involved after long-term administration.

The anti-G-CSF antibody species (monoclonal rat IgG₁ clone) is now provided on p.22. Only a short-term treatment duration of 3-4 weeks was approved by the Walter and Eliza Hall Animal Ethics Committee, so for this purpose allo-reactivity was not an experimental concern.

4. Irradiation dose seems high.

The primary aim of the bone marrow reconstitution experiment was to clearly distinguish between radioresistant and radiosensitive cells, rather than assessing a clinically relevant dose in our preclinical model. Overall irradiation was well tolerated, and reconstitution failure rate was very low.

5. They need to distinguish skin neutrophils from neutrophils in the microcirculation (ie. Blood neutrophils in skin sections).

Thank you very much for raising this point. We now use CD31⁺ to stain for endothelial cells and MPO⁺ as previously to distinguish blood neutrophils from tissue neutrophils (p.6 and Supp.fig.1d).

6. Are lung neutrophils including BALF? Was the lung perfused prior to dissociation and FACS analysis?

BALF has not been included in further analysis. Mice were perfused prior to dissociation and FACS analysis, which is now stated on p.19.

Results

Clear presentation of data but there are some errors

Fig 1.

Fig 1i,j,l,m – were lung counts significant?

Yes, neutrophil counts in the lung during disease onset were significantly different ($p < 0.0453$) which is now outlined now in Fig.1i. In contrast, lung counts in Fig.1j, l, m did not reach levels of significance.

Fig 1P – I don't see the reduction in Igs. If anything IgG 2a is increased.

We agree and have changed the wording accordingly (in the text on p.7 and in the figure legend on p.31).

Fig 2.

Fig 2 b shows significant differences between KI and KI on IL-6 but TNF does not show significance but this is not stated.

We added a sentence with regards to bodyweight of $PLC\gamma 2^{S707Y/+}TNF^{-/-}$ mice (p.7-8).

Fig 2 f again I do not see the Ig differences.

Only the Ig2a levels differ between subgroups, which is now stated on p.8.

Fig 2 g- what is going on with IL-12p40 in mouse

This is an interesting point. IL-12p40 is the shared subunit composing both IL-12 (with IL-12p35 forming IL-12p70) and IL-23 (with IL-12p19). We did not see a significant increase in IL-12p70, and although we did not quantify IL-23 directly, it typically induces IL-17, which was also not upregulated in this analysis of the APLAID mice. Nevertheless, IL-12p40 may contribute to the mobilisation of haematopoietic progenitors (Jackson JD, Blood 1995, PMID: 7727771), IgG2a production by mouse B cells (Rodolfo M, Cancer Res 1998, PMID: 9865740) and skin pathology (Toichi E, JI, PMID: 16982934). For all the reasons above IL-12p40 may play some role in disease pathology of our preclinical APLAID mouse model.

Fig 3

Cytokine heat maps should show individual data points including the limit of detection.

We changed Fig.3 according to the reviewer's suggestions making the individual data points and limit of detection clear.

Consider moving supp Fig 3 into manuscript- comparisons to different autoinflammatory diseases is novel.

Although we are also enthusiastic about these comparisons, it remains a preliminary dataset with small numbers of patients. Given these limitations, and in response to referee 2, we would suggest retaining this figure in the supplementary data.

What is going on with IL-12p70 and IL-1b in human

Unlike in mice, both IL-12p40 and IL-12p70 were upregulated in APLAID patients. It remains possible that these cytokines play some role in disease pathology, however IL-12p70 is typically considered to be anti-inflammatory in the skin (Kulig et al. Nat Commun 2016, PMID: 27892456), suggesting that it may not be a dominant cytokine in this disease.

Increased levels of IL-1b might be the consequence of Ca²⁺ flux following PLCg2 activation, which in turn may activate the NLRP3 inflammasome, thus resulting in IL1b release. However, we suspect that this is elevated along with other non-specific inflammatory markers in APLAID patients, and that it is unlikely to play a significant role in disease given the lack of response to anakinra, and data from our mouse model.

Fig 4

Fig 4f. Igs are decreased with treatment.

This is now stated on p. 10 and in the figure legend of figure 4 on p.32.

Fig 5 and Fig 6 could be combined

We agree to combine fig.5 and fig.6, a point that was also raised by referee #2.

Fig 7

Obvious experiment is to do the opposite BMT control – Mutant BM into WT

This is an important suggestion. We performed the suggested experiment (Mutant BM into WT), and also transplanted mutant bone marrow into G-CSF^{-/-} mice to assess, if the cytokine is derived from the haematopoietic or stromal compartment. All data are shown in fig.7.

Fig 7e appears to be a mistake - legend is switched? Colours should match the rest of the figure.

Thank you for pointing out this mistake. We changed the colours of the symbols accordingly. The former fig.7 is now fig.6.

Supplementary Figures

Mostly appropriate placement of supplemental data

Supp Fig 6. The FACS data needs compensation adjustment - cells cannot be sitting on the axis. Also needs a different format to convey how many cells are in these plots.

Current format is hard to interpret.

Supp Fig 6. The gating proposes a population of progenitors without the use of a lineage gate. This is essential. The gating for cKIT relative to the 'non-myeloid' population appears arbitrary and inappropriately set. The same is true for the Sca1 gate - are the authors claiming this entire population is Sca1 positive? Clear definitions of the progenitors needed here and should conform to the literature in this regard unless otherwise justified.

We have amended the FACS data with minor correction on compensation. Data are now shown in pseudocolor format to reveal off-scale events. Ticks and numbers are also provided in the adapted Suppl.fig.6.

Shown FACS data are representative of BM cells from an APLAID mouse.

We did not use conventional Lin⁻ strategy as we wish to identify neutrophils and monocytes at various stages of maturation. Following the exclusion of cell debris, doublets and dead cells, we identified mature neutrophils as Ly6G⁺CD11b⁺ cells, which only have ~8% BrdU⁺ cells; in agreement with their terminally differentiated status prior to entry to the circulation. In the Ly6G⁻ fraction, we identified all monocytic lineage cells as CD115⁺ SSC^{low} cells, which include CD11b⁺ low proliferating monocyte/committed monocyte progenitor and CD11b⁺ monocytes. We subsequently gated on CD115⁻ SSC^{low} cells to identify CD16/32⁺ myeloid progenitor cells, excluding CD16/32⁻ megakaryocyte-erythrocyte progenitor (MEP) and common lymphocyte progenitor (CLP) (Chapple *et al*, Blood Advances 2018, PMID: **29848758**). CD16/32⁺ cells were further gated as Sca1⁻ cells to exclude contaminating HSCs and other early progenitors. As we did not include newer markers such as CD81 or CD106 for neutrophil progenitors (NeuP; Kwok Immunity 2020, PMID: 32579887), we defined myeloid progenitors (MyeloP) as CD11b⁻ cKit⁺ MyeloP or CD11b⁺ c-Kit⁻ MyeloP.

CD11b⁺ c-Kit⁻ MyeloP were further distinguished as Ly6C^{hi} and Ly6C^{int} MyeloP subsets. We defined Ly6C^{int} cells as NeuP given their higher granularity/SSC relative to Ly6C^{hi} subset.

Discussion

Data supports conclusions

Fonts are messed up

This section is written in a very choppy/sloppy manner unlike the rest of the manuscript

We harmonized the different fonts and re-wrote the discussion extensively (p.14-17).

References - Adequate

Reviewer #2:

Remarks to the Author:

The manuscript by Mulazzani et al. describes novel findings that substantially expand our understanding of development of APLAID – dominantly inherited, monogenic immune disorder. Furthermore, the data suggest new therapeutic options linked to the high levels of G-CSF as a critical mediator in this pathology.

The evidence is comprehensive and experimental approaches and results appear to be without major problems. Some additional data and extensive improvements of the manuscript text are needed to further support and clearly present the findings and, importantly, to outline their wider relevance.

Specific points of criticism:

(1) Background information presented in Introduction should be improved. Specific points are below.

- Only 2 clinical studies for APLAID are referenced (ref 1 and 2). Because the number of all reported APLAID cases is relatively small - all should be referenced. Adapting and updating Table 1 in reference 2, as a supplemental information, is needed for the understanding of APLAID by broader readership.

All reported APLAID cases are now referenced (p.3). A supplementary table (Table 1S) summarizing all published APLAID cases is now provided.

- Signalling context of PLC γ 2 is not well explained. Notably - BLNK is not a tyrosine kinase. Statements should also be more specific about the cell types and receptors and distinguish B cells from other types. References 9, 10 and 11 are outdated; there are more recent relevant reviews focused on PLC. Similarly, description of the 3D structural organization should be based on more recent direct studies of PLC γ 1 structure and models of PLC γ 2 structure (see AF-P16885-F1), not ref 12.

The signalling context of PLC γ 2 is now expanded and references are updated with more recently published reviews (p. 4).

- Introduction to the previously proposed link to NLRP3 should better cover and explain data in refs 2 and 13.

The introduction now covers the possible link to NLRP3 and explains data demonstrated in ref 2 and 13 (p. 4).

- The authors should briefly mention information gained from studies described in refs 14 and 15, not just the limitations of this work.

Yes, our mechanistic understanding on the role of PLC γ 2 in perpetuating autoinflammation has been greatly improved by these studies (ref. 14 and 15). Major study outcomes are now highlighted in the introduction (p. 4 and 5).

(2) Figure 1 presents comprehensive analyses of PLC γ 2S707Y/+ mice. It is also stated that “the clinical phenotype of PLC γ 2S707Y/+ mice strongly recapitulates the human disease”. Some further clarification of aspects that may not be “strongly recapitulated” should be provided. For example, are the findings shown in Figure 1p expected, based on data from patients?

We appreciate this comment and have updated the text accordingly. Specifically, the main clinical and immunological differences of PLC γ 2^{S707/+} mice compared to APLAID patients include a lack of inflammatory eye diseases and a lack of immunoglobulin reduction, which is now stated on page 7.

(3) Description of the data obtained from crosses resulting in PLC γ 2S707Y/+IL-6-/-, PLC γ 2S707Y/+caspase-1-/- and PLC γ 2S707Y/+TNF-/- mice, needs further clarification. Specifically, know features of the phenotypes of the KO mice crossed with PLC γ 2S707Y/+ mice should be included as a supplemental table.

An additional table now covers the phenotype of the different PLC γ 2 intercrosses and is provided as supplementary table (Table S2).

(4) The author’s state: “Multiplex cytokine assessment of skin and lung lysates also revealed increased G-CSF and MIP-1 α/β in PLC γ 2S707Y/+mice, but not in PLC γ 2S707Y/+IL-6-/-, PLC γ 2S707Y/+TNF-/- and PLC γ 2S707Y/+caspase-1-/- mice (Supp.Fig.2f-g)”. Considering the importance of G-CSF highlighted in this study, some further explanation should be provided about the implication of this difference between PLC γ 2S707Y/+mice and the crosses.

In line with this suggestion, we now discuss local and systemic effects of G-CSF in the context of the different knock-out strains. Information is given in the discussion p. 16.

(5) The main conclusion from the genetic approach rules out the key importance of IL-6, inflammasome or TNF in driving APLAID. Considering that the evidence for the role of G-CSF has been obtained using pharmacological tools, it would be helpful to have a similar information that complements the genetic studies; specifically, the effect on

PLCy2S707Y/+ mice of relevant pharmacological agents used in the clinic that block IL-1, JAK1/2 or TNF. These should be addressed experimentally. Conversely, it would be informative to know the phenotype of G-CSF KO mice.

Although pharmacologic inhibition of IL-1, JAK1/2 or TNF would add information, it is likely to be negative data given the minimal patient responses to all these approaches, and therefore we could not justify the expenditure of resources and mouse lives. We also did not perform constitutive deletion of G-CSF due to time constraints, however we did generate chimeric mice with APLAID mutant bone marrow and G-CSF sufficient or deficient non-haematopoietic cells, which was highly informative (new Fig 7). Specifically, this experiment demonstrates that the APLAID mutation is active in haematopoietic cells, and signals to non-haematopoietic cells, which then make pathogenic G-CSF.

(6) The information related to 2 clinical cases in Figure 3 doesn't provide a direct or conclusive support for the key role of G-CSF in APLAID. Nevertheless, this information is interesting but should be better explained and discussed, in particular, the implications of the high G-CSF not being unique for APLAID monogenic disease or even less prominent compared to other immune disorders. Related to this, it is not clear, based on a small number of cases for each disorder, what are the differences between the individuals with the same disorder compared to different disorders.

Yes, we appreciate that there are limitations of this preliminary study and have now updated figure 3 to clearly identify the individual data points and limit of detection of the assay. The lack of specificity for G-CSF elevations in APLAID compared with other autoinflammatory syndromes is also now discussed (p. 15) and a detailed overview of all patients is given in Table S1.

(7) Data shown in Figures 5 and 6 and the related text could be better linked under one subheading in the result section.

We agree to link Fig 5 and Fig 6 together, a point that was also raised by referee #1.

(8) Discussion should be improved and include/elaborate some of the points.

- The authors should better connect known and new observations into possible steps of the disease development, linking cellular and organ/tissue levels.
- The role of G-CSF in non-monogenic inflammatory diseases should be explained and any parallels with APLAID highlighted.

Thank you very much for raising this point. We re-wrote the discussion extensively (page 14-17), which was also requested by referee #1.

Decision Letter, first revision:

9th Jan 2023

Dear Dr. Masters,

Thank you for submitting your revised manuscript "G-CSF drives autoinflammation in APLAID" (NI-A34612A). It has now been seen by the original referees and their comments are below. The reviewers find that the paper has improved in revision, and therefore we'll be happy in principle to publish it in Nature Immunology, pending minor revisions to satisfy the referees' final requests and to comply with our editorial and formatting guidelines.

We will now perform detailed checks on your paper and will send you a checklist detailing our editorial and formatting requirements in about a week. Please do not upload the final materials and make any revisions until you receive this additional information from us.

If you had not uploaded a Word file for the current version of the manuscript, we will need one before beginning the editing process; please email that to immunology@us.nature.com at your earliest convenience.

Thank you again for your interest in Nature Immunology Please do not hesitate to contact me if you have any questions.

Sincerely,

Nick Bernard, PhD
Senior Editor
Nature Immunology

Reviewer #1 (Remarks to the Author):

The authors have adequately addressed my concerns

Reviewer #2 (Remarks to the Author):

The revised manuscript by Mulazzani et al. addresses most points of criticism and incorporates related changes in the text and figures. However, this very substantially improved version contains some mistakes and lacks clarity in some important sections in Discussion. Specific comments are below.

- When describing a model covering the link between PLCg2 APLAID variant and G-CSF, the sentences on page 16, lines 372-374 and page 17, lines 386-388 need to be stated clearly. Furthermore, a

comparison between an elusive “soluble factor” and the research compound OAG may not be appropriate.

To better present an overall mechanistic hypothesis, a diagram or other graphical illustration can be included as a supplemental figure.

- Sentence on page 3, lines 68-70 should be corrected to reflect the following: Two interaction surfaces (the split PH/catalytic domain and the cSH2/C2 domain) keep PLCg2 in an autoinhibited form.
- Statement on page 4, line 74, should be changed to “non-receptor” tyrosine...
- Typographical errors, for example line 84 “celltype”
- Font in Discussion is not consistent

Author Rebuttal, first revision:

Response to Reviewer Comments

Reviewer #1 (Remarks to the Author):

The authors have adequately addressed my concerns

Reviewer #2 (Remarks to the Author):

The revised manuscript by Mulazzani et al. addresses most points of criticism and incorporates related changes in the text and figures. However, this very substantially improved version contains some mistakes and lacks clarity in some important sections in Discussion. Specific comments are below.

- When describing a model covering the link between PLCg2 APLAID variant and G-CSF, the sentences on page 16, lines 372-374 and page 17, lines 386-388 need to be stated clearly. Furthermore, a comparison between an elusive “soluble factor” and the research compound OAG may not be appropriate.

We toned down the indicated parts of the discussion changed the sentences according to the reviewer’s suggestions.

To better present an overall mechanistic hypothesis, a diagram or other graphical illustration can be included as a supplemental figure.

A graphical scheme is now provided as supp.fig.8b.

- Sentence on page 3, lines 68-70 should be corrected to reflect the following: Two

interaction surfaces (the split PH/catalytic domain and the cSH2/C2 domain) keep PLCg2 in an autoinhibited form.

We changed the sentences according to the reviewer's suggestions.

- Statement on page 4, line 74, should be changed to "non-receptor" tyrosine...

We changed tyrosine kinase to "non-receptor" tyrosine kinase.

- Typographical errors, for example line 84 "celltype"

We corrected typographical errors accordingly.

- Font in Discussion is not consistent

Thank you very much for raising this point. We harmonized fonts in the discussion.

Final Decision Letter:

In reply please quote: NI-A34612B

Dear Dr. Masters,

I am delighted to accept your manuscript entitled "G-CSF drives autoinflammation in APLAID" for publication in an upcoming issue of Nature Immunology.

Over the next few weeks, your paper will be copyedited to ensure that it conforms to Nature Immunology style. Once your paper is typeset, you will receive an email with a link to choose the appropriate publishing options for your paper and our Author Services team will be in touch regarding any additional information that may be required.

Please note that *Nature Immunology* is a Transformative Journal (TJ). Authors may publish their research with us through the traditional subscription access route or make their paper immediately open access through payment of an article-processing charge (APC). Authors will not be required to make a final decision about access to their article until it has been accepted. [Find out more about Transformative Journals](https://www.springernature.com/gp/open-research/transformative-journals).

Your paper will be published online soon after we receive your corrections and will appear in print in the next available issue. Content is published online weekly on Mondays and Thursdays, and the embargo is set at 16:00 London time (GMT)/11:00 am US Eastern time (EST) on the day of publication. Now is the time to inform your Public Relations or Press Office about your paper, as they might be interested in promoting its publication. This will allow them time to prepare an accurate and satisfactory press release. Include your manuscript tracking number (NI-A34612B) and the name of the journal, which they will need when they contact our office.

About one week before your paper is published online, we shall be distributing a press release to news organizations worldwide, which may very well include details of your work. We are happy for your institution or funding agency to prepare its own press release, but it must mention the embargo date and Nature Immunology. Our Press Office will contact you closer to the time of publication, but if you or your Press Office have any enquiries in the meantime, please contact press@nature.com.

Also, if you have any spectacular or outstanding figures or graphics associated with your manuscript - though not necessarily included with your submission - we'd be delighted to consider them as candidates for our cover. Simply send an electronic version (accompanied by a hard copy) to us with a possible cover caption enclosed.

Please note that we encourage the authors to self-archive their manuscript (the accepted version before copy editing) in their institutional repository, and in their funders' archives, six months after publication. Nature Portfolio recognizes the efforts of funding bodies to increase access of the research they fund, and strongly encourages authors to participate in such efforts. For information about our editorial policy, including license agreement and author copyright, please visit www.nature.com/ni/about/ed_policies/index.html

An online order form for reprints of your paper is available at <https://www.nature.com/reprints/author-reprints.html>. Please let your coauthors

and your institutions' public affairs office know that they are also welcome to order reprints by this method.

Sincerely,

Nick Bernard, PhD
Senior Editor
Nature Immunology